# Antibiotic export by MexB multidrug efflux transporter is allosterically controlled by a MexA-OprM chaperone-like complex

Marie Glavier [1,2,7], Dhenesh Puvanendran [3,4,7], Dimitri Salvador[1,2,7], Marion Decossas[1,2], Gilles Phan [5], Cyril Garnier[5], Elisa Frezza [5], Quentin Cece[3,4], Guy Schoehn [6], Martin Picard[3,4], Jean-Christophe Taveau[1,2], Laetitia Daury[1,2,8], Isabelle Broutin [5,8 ✉] & Olivier Lambert [1,2,8 ✉]

The tripartite multidrug efflux system MexAB-OprM is a major actor in *Pseudomonas aeruginosa* antibiotic resistance by exporting a large variety of antimicrobial compounds. Crystal structures of MexB and of its *Escherichia coli* homolog AcrB had revealed asymmetric trimers depicting a directional drug pathway by a conformational interconversion (from Loose and Tight binding pockets to Open gate (LTO) for drug exit). It remains unclear how MexB acquires its LTO form. Here by performing functional and cryo-EM structural investigations of MexB at various stages of the assembly process, we unveil that MexB inserted in lipid membrane is not set for active transport because it displays an inactive LTC form with a Closed exit gate. In the tripartite complex, OprM and MexA form a corset-like platform that converts MexB into the active form. Our findings shed new light on the resistance nodulation cell division (RND) cognate partners which act as allosteric factors eliciting the functional drug extrusion.

[1] University of Bordeaux, CBMN UMR 5248, Bordeaux INP, F-33600 Pessac, France. [2] CNRS, CBMN UMR5248, F-33600 Pessac, France. [3] Laboratoire de Biologie Physico-Chimique des Protéines Membranaires, Université de Paris, UMR 7099 CNRS, F-75005 Paris, France. [4] Institut de Biologie Physico-Chimique, F-75005 Paris, France. [5] Laboratoire CiTCoM, Université de Paris, CNRS, 75006 Paris, France. [6] Université Grenoble Alpes, CNRS, CEA, Institute for Structural Biology (IBS), 38000 Grenoble, France. [7] These authors contributed equally: Marie Glavier, Dhenesh Puvanendran, Dimitri Salvador. [8] These authors jointly supervised this work: Laetitia Daury, Isabelle Broutin, Olivier Lambert. ✉email: isabelle.broutin@parisdescartes.fr; o.lambert@cbmn.u-bordeaux.fr

In Gram-negative bacteria, tripartite multidrug efflux systems export biological metabolites and antimicrobial compounds, thereby contributing to bacterial resistance, which emerges as a major health concern[1,2]. The understanding of the functional mechanism of these exporting systems is a prerequisite in the development of blockers that would restore the efficiency of the existing therapeutic arsenal, focusing the interest of intense research[3]. Tripartite systems of the Resistance Nodulation cell Division (RND) superfamily are composed of an inner-membrane RND transporter driven by the proton motive force, an outer-membrane factor (OMF), and a periplasmic membrane fusion protein (MFP), forming a tripartite pump with a contiguous exit duct[4–6]. Archetypical RND transporters operate as a secondary proton/drug antiporter exploiting the energy associated to proton transport in their transmembrane domains to power the export of substrates through the periplasmic pore domain, 60 Å away, via long-range conformational communication[7,8]. Among the major RND actors in drug resistance in *Pseudomonas aeruginosa*, MexB transporter works in conjunction with OprM (OMF component), and MexA (MFP component). Crystal structure determination revealed that MexB as well as the *E coli* homologous AcrB are asymmetric homotrimers for which the three monomer conformations representing consecutive states were designated Loose (or Access), Tight (or Binding), and Open (or Extrusion)[9–12]. A hydrophobic pocket created in the T monomer was described as a substrate binding pocket based on the structures of AcrB-drug complexes showing the binding of minocyclin, and doxorubicin to this pocket[9]. From the asymmetric AcrB structures, a model for a directional drug transport based on conformational cycling of the monomers (referred also as functionally rotating mechanism) by utilizing proton binding and dissociation has been proposed with the substrate entry happening in the L monomer, followed by a conformational change to the T monomer resulting in the substrate binding to the binding pocket and finally the substrate release arose after T monomer conversion to the O monomer. The cycling event was regenerated after the conversion of the O monomer to the L monomer. This model has been supported by additional experimental studies[13,14], molecular dynamics simulations[15–17], and thermodynamic calculations[18,19]. As a result of these studies, specific substrate recognition sites have been identified based on AcrB structure. The access pocket is placed in the vestibule region between PC1 and PC2 subdomains and open in protomers L and T[20,21]. The distal pocket separated from the access pocket by the switch loop is located more closely to the funnel domain and open only in the T protomer[20,21]. The exit gate formed by residues Gln124, Gln125 (part of the later called helix gate corresponding to the Nα2′ helix[22]), and Tyr758 gives access to the central funnel at the top of the periplasmic domain and is only open in the O protomer[11,15,16].

Intriguingly symmetric trimers have been reported for AcrB[22] and for other RND transporters[23–25] i.e CmeB[23], CusA[24], ZneA[26], AdeB[27]. Single-molecule FRET studies of CmeAB in proteoliposome have suggested that each protomer experiences independent conformational changes[23] maintaining some uncertainty on the direction of the monomer conversion (e.g. from L to T, T to O, and O to L) and on a common cycling mechanism of RND drug transport.

Recent cryo-Electron Microscopy (cryo-EM) studies of AcrAB-TolC and MexAB-OprM systems stabilized with amphiphilic polymer have shown overall similar architectures of six MFP surrounding a trimer of RND and in a tip-to-tip interaction with the OMF. Therein, AcrB and MexB adopt an asymmetric LTO conformation in a subset of particles[28,29]. Interestingly, in both studies tripartite structures have been determined in the presence and in the absence of substrate describing a transport-state and a resting state respectively. According to Wang et al.[28], the resting state corresponding to the apo form with TolC in closed-state and AcrB in symmetric conformation switched to a transport-state in the presence of puromycin in which the AcrB trimer adopted asymmetric conformations and TolC is open. For Tsutsumi et al.[29], the resting state corresponded to the tripartite structure with an opened OprM and an asymmetric MexB trimer as in crystal LTO conformation except for the gate loop in the T protomer. In the presence of novobiocin, T protomer underwent a slight conformation change. The gate loop was stretched and descended allowing novobiocin binding in the distal-binding pocket. From all the above, although it has been postulated that the LLL to LTO transition could be driven upon substrate binding[27,28,30,31], it remains highly debated whether the LTO form preexists as the resting state[32], or if it is acquired at some point during the transport process. However, as far as we are aware of, a scenario where binding of the cognate partners is the driving factor for the elicitation of transport has, so far, never been considered.

In complement to structural analysis, quantitative measurement of substrate transport is crucial for the detailed understanding of efflux pumps. In 2005, Aires and Nikaido discussed their results regarding the activity of AcrD into proteoliposome and concluded that ~0.3 proton are consumed per trimer of AcrD and per second[33]. In another study, EtBr transport by MexB gave rise to an impressive turnover rate of $500\,s^{-1}$ based on a number of pumps previously estimated by immunoblotting methods[34,35]. Hence, the exact order of magnitude of the velocity of transport is still debated. We have decided to address this issue taking advantage of a methodology allowing the monitoring of efflux pumps activity after reconstitution into proteoliposomes and monitoring proton transport using a pH-sensitive fluorescent probe. Functional assay using MexAB-OprM system reconstituted in liposomes has shown an efficient drug transport correlated with a proton gradient[36,37]. Hence, activity of RND efflux pumps can be assessed by monitoring their substrate-induced proton consumption. A correlation of transport activity of MexB at various stages of the assembly process with its structure solved in lipid environment using nanodisc (ND) methodology[38] should provide an integrated view.

Here, by correlating functional and cryo-EM structural investigations of MexB trimer alone in lipid membrane (termed MexB-solo) and associated in the tripartite complex (termed MexB-trio as opposed to MexB-solo), we reveal that MexB-solo displays almost no activity and exhibits a new asymmetric conformation revealing that the exit pathway of the protomer involved in drug release is now closed depicting a LTC conformation. Unlike MexB-solo, MexB-trio is fully active and shows the LTO conformations. Conversion of MexB to the active LTO form is mediated by OprM and MexA forming a tight corset-like platform giving rise to the required conformational changes.

## Results

**Functional transport of MexB during the tripartite assembly process.** Previously, we measured the activity of the MexAB-OprM pump in a system that lacked the temporal resolution necessary to accurately monitor rates of transport[37]. The acceleration of acidification upon transport happened during a "blind window" of few seconds necessary to acidify the fluorescence cuvette in a standard fluorometer. We have adapted our methodology to stopped-flow fast recording in order to compare transport activity of MexB at several stages of the assembly process of the pump i.e. MexB alone (MexB-solo), MexB associated with MexA and MexB engaged in the MexAB-OprM complex (MexB-trio) (Fig. 1a). Proteins reconstituted into liposomes were rapidly subjected to acidification. We have improved

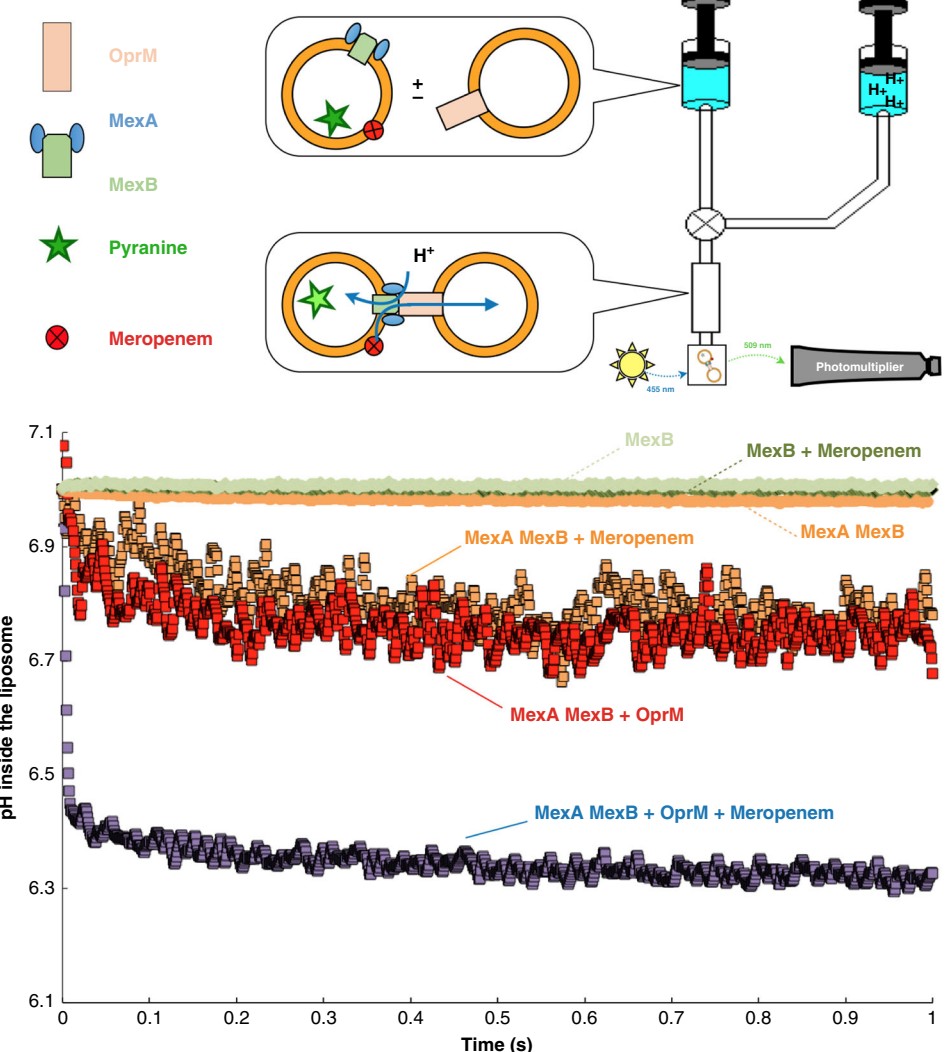

**Fig. 1 Transport activity. a** Schematic representation of the stopped-flow measurement. **b** Kinetics of acidification of proteoliposomes in the absence ("MexB", light green; "MexA MexB", light orange; and "MexAMexB + OprM", red curves) or in the presence of 5 µM meropenem ("MexB", dark green; "MexA MexB", orange; and "MexAMexB + OprM", purple curves). The transporter is present under its wild type form in the presence or in the absence of the MexA partner. Proteoliposomes were incubated for 15 min with meropenem and then subjected to rapid mixing with the acidification buffer (traces are cumulative averages of 5–10 measurements for each batch of reconstituted protein). Measurements in the presence of OprM proteoliposomes (red and purple curves) were performed under the same conditions described above, except that OprM proteoliposomes were incubated for at least one hour before the addition of meropenem.

our previously described protocol[37] using a stopped-flow apparatus in order to record transport measurements on a millisecond time scale. A small, but reproducible, meropenem-induced acidification is mediated by MexB wild-type in the presence of MexA (Fig. 1b, orange curve) while in the absence of MexA almost no proton transport is measured for MexB-solo despite the presence of substrate as observed for several RNDs[4,33,36] (Fig. 1b, light green, dark green and light orange curves). This liposome acidification is strictly related to proton transport by MexB as confirmed with the inability of an inactive MexB$_{D407N}$ mutant[39] to convert the proton gradient into actual transport (Supplementary Fig. 1, white curve). Most interestingly when MexB has been preassembled with MexA and OprM a liposome acidification is observed (Fig. 1b, red curve) and the rate of proton transport is dramatically increased in the presence of meropenem (Fig. 1b, purple curve) indicating that, MexB-trio is functionally active unlike MexB-solo. Thanks to the adaptation of our methodology to stopped-flow fast recording, we show that proton transport

turns out to be an extremely fast event: with a $t_{1/2}$ of ~100 ms, reflecting that MexB is a rapid and efficient machine when assembled with its partners.

**Cryo-EM structures of MexB-solo and MexB-trio**. To get a molecular explanation for this effect, we have investigated the conformation of MexB trimer before and after being assembled in tripartite complex. We reconstituted MexB in a lipid environment using lipid nanodiscs and mixed with MexA and OprM also stabilized in nanodisc leading to a self-assembled tripartite complex MexAB-OprM[38]. Among the particles identified on the cryo-EM grid, tripartite complexes and isolated components of the pump were visualized. Hence structures of tripartite complexes as well as individual MexB particles (MexB-solo), originating from the same protein batch, have been determined by single-particle cryo-EM from the same set of micrographs (Supplementary Fig. 2).

Firstly, the 3.2 Å resolution tripartite complex structure in nanodisc revealed a 1:2:1 stoichiometry of OprM, MexA, and MexB with a OprM trimer, a MexB trimer (MexB-trio) and a MexA hexamer surrounding MexB and interacting with OprM in an open conformation. The α-hairpin domain of MexA contacts the extremity of the periplasmic helices of OprM while the β-barrel domain and the membrane proximal (MP) domain interact with the funnel domain and the pore domain of MexB respectively. The overall structure was similar to that of MexAB-OprM stabilized in amphipol[29]. Likewise, two complexes geometrically related by a 60° rotation of either OprM or MexB were identified (Supplementary Fig. 2f), suggesting two binding modes of OprM to MexA[29]. MexB-trio is surrounded by six molecules of MexA anchored in the lipid membrane. The densities protruding from the MexA-MP domain unveiled the N-terminal segment extending down to the lipid membrane (Supplementary Fig. 3). This 12 amino-acids tail that has not been described before[29,40] formed a stretch segment attached to the lipid membrane. The lipid moiety of the N-terminal cysteine could not be resolved because it is embedded inside the phospholipid layer density. MexB-trio adopted an asymmetric LTO conformation for all the particles contributing to the cryo-EM 3D reconstruction (Fig. 2 and Supplementary Fig. 4). These conformational states were similar, notably in the pore domain (Supplementary Fig. 3), to those of the crystal and cryoEM structures in the absence of substrate[12,29], which is in perfect agreement with the functional rotating mechanism[32].

Secondly, isolated MexB particles (MexB-solo) were processed independently of the tripartite complex. The resulting reconstruction yielded a density map at ~4.6 Å resolution (Fig. 2a and Supplementary Fig. 5) revealing an asymmetric conformation of MexB-solo different from that of MexB-trio (Fig. 2b). In MexB-solo, the protomer L resembled the one in MexB-trio (rmsd on 1030 Cα 1.11 Å) while the two others showed notable conformational differences in the pore domain. Structural deviation (rmsd 1.43 Å) of the related protomer T mostly concerned the peripheral of the domain PN2, so that the internal features of the drug pathway, including the deep binding pocket, remained the same. By contrast, in the third protomer of MexB-solo which is expected to be involved in drug release as observed in the O protomer of MexB-trio, the most striking change (rmsd 1.43 Å) concerned the central helix plus helix gate (amino acids 99–124) (Fig. 2c, d). We described a new protomer conformation named C for "Closed" because it hampers the exit of the drugs in MexB-solo compared to the open form O in MexB-trio. Overall, MexB-solo adopted a LTC conformation while MexB-trio is in the LTO conformation, and structural comparisons revealed differences mainly in PN2 subdomain of protomer T, and in PC2, PN1 subdomains and the central helix plus helix gate of protomer C/O (Fig. 2b–d and Supplementary Fig. 8).

**Rigid scaffold of MexA induces LTO conformation of MexB.** In the tripartite complex, six MexA protomers formed a symmetric dome on top of MexB with their lipoyl and hairpin domains, and used their β-barrel and MP domains for interacting with MexB subdomains. This MexA hexameric dome resulted from the arrangement of two non-equivalent MexA molecules (MexA-I and MexA-II) bound *per* MexB protomer (Supplementary Fig. 3)[28]. For MexA-I, the β-barrel and MP domains interacted respectively with the β-hairpin loop (DC) and the turn of the long β–loop (DN) crossing each monomer of MexB respectively without major overall structural modifications. Unlike MexA-I, the MexA-II contacts were at the vicinity of the conformational changes observed on MexB-trio (Figs. 2b, c and 3). The β-barrel

domain of MexA-II pinched by two contact sites (R34 and T233) the β-hairpin loop (DN) of MexB which was pushed away by more than 6 Å compared to its position into MexB-solo indicating a key conformational change to accommodate the assembly (Fig. 2b) as described for the homologue efflux pump in *E. coli* AcrAB-TolC[28]. Interestingly the deletion of the same β–hairpin loop (Δ252–256) in AcrB resulted in an increase in drug sensitivity, emphasizing the importance of this contact for drug efflux[41]. In addition to these stabilizing contacts, MexA-II MP was in interaction with PN2 domain of the protomer T which exhibited a conformational change regarding protomer T of MexB-solo (Fig. 2b, c). Comparatively, PN2 domain of MexB-solo was geometrically too close to MexA-II MP generating steric hindrances that are sufficient to induce the observed shift of PN2 (Fig. 3). The constraint on the PN2 domain of MexB protomer T at residue Q319 imposed by MexA-II MP created the contact surface which is unique compared with those between MexA-II MP and protomers L and O (Supplementary Fig. 6). It is of interest to note that the mutation of the equivalent residue in AcrB (S319C) resulted in a loss of resistance to novobiocin[42]. Therefore, by interacting with PN2 subdomain of protomer T (T-PN2), MexA-II-MP triggered cascading movements inducing the movement of T-PN2 which was in turn transmitted to the neighboring domains of protomer C including C-PC2 shifting towards C-PC1 (loss of C-PC2 stabilization by T-PN2), thereby opening the exit pathway to the funnel domain by tilting away the C-PN1 helix gate (Fig. 2b–d and Supplementary Fig. 7 and Supplementary Video).

**OprM-MexA and MexA-MexB interfaces governed by two drastically different energetic levels.** Before being assembled in the complex, the LTC conformation of MexB-solo prevents drug export (Supplementary Fig. 7). Accordingly, as shown by the activity assays, MexB acquires its active form only when assembled with MexA and OprM leveraging the two protein-protein interfaces OprM-MexA and MexA-MexB. A comparison of solvent accessibility, binding energy and electrostatic potential calculated from the Poisson-Boltzmann equation has been performed to gain a better understanding of the principles governing the interactions involved in the OprM-MexA and MexA-MexB interfaces. For OprM-MexA, we observed a complementarity of the electrostatic potential energy that was not present in the MexA-MexB interface (Fig. 4 and Supplementary Fig. 9). Furthermore, by calculating the accessible surface area and the gap volume, these two interfaces had a very different gap index resulting in a high complementarity for OprM-MexA interface and a lower complementarity for MexA-MexB (Supplementary Table 1). A simplified estimation of the binding energy at the two interfaces suggested a stronger interaction between OprM and MexA than between MexA and MexB (Supplementary Table 2). For OprM-MexA interface, the six helical hairpins of MexA established a strong and stable tip-to tip interaction with the six helix-turn-helices formed by OprM trimer in an open conformation (in agreement with molecular dynamic[43]). In contrast to the OprM-MexA interface, the MP and β-barrel domains of MexA formed a loose interface, with a reduced energy binding, resulting in a cap on top of the MexB trimer that would allow its required alternative movements. Overall, we show that the functional assembly is mediated by a scaffolding chaperon effect of MexA on MexB via OprM. MexA constrained the conformation of MexB protomer T contributing to switch the neighboring protomer in an open conformation which is not energetically-favored in MexB-solo, as illustrated by its inability to achieve any proton transport (Fig. 1b, light green and dark green curves).

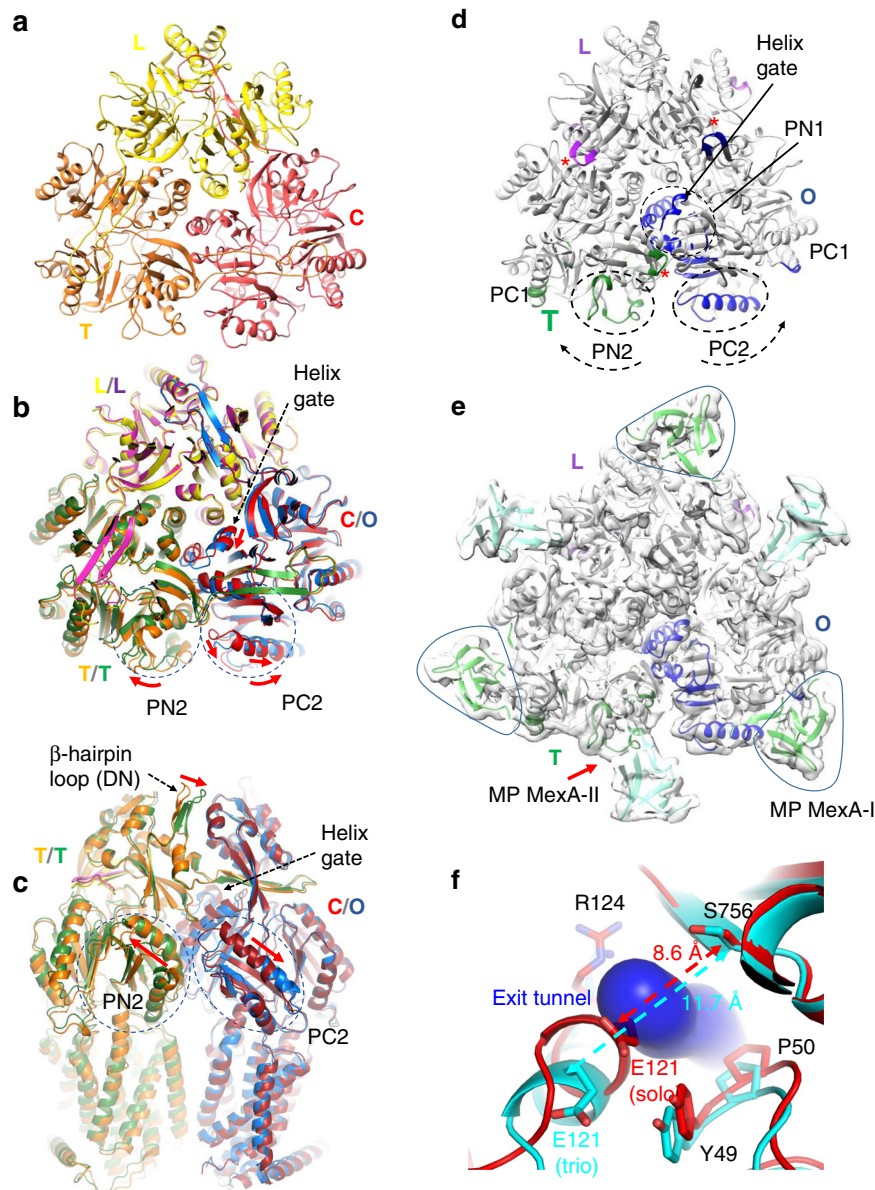

**Fig. 2 Structure of MexB-solo and MexB-trio. a** Pore domain visualization of MexB-solo trimer in LTC conformation shown in yellow, orange and red for the L, T, and C protomers respectively. **b**, **c** Superimposition of MexB-solo and MexB-trio (purple, green, and blue for the L, T, and O protomers respectively) viewed from the top (**b**) as in **a** and perpendicular to the membrane (**c**). In the periplasmic part, main conformational changes observed between MexB-solo and MexB-trio occurred in the three β-hairpin loops (DN domain) (**c**), in the T protomer PN2 domain, and in the neighboring protomer switched from C to O state with a combined movement of the PC2 domain and the central helix plus helix gate (PN1 domain). The described movements are illustrated by red arrows. **d** Pore domain visualization of MexB-trio trimer. The conformational changes observed in **b**–**c** between MexB-solo and MexB-trio are shown in purple, green and blue (for the L, T, and O protomers respectively), according to the root mean square deviations. Parts of the structure with rmsd lower than 2 Å are in gray. The PN2 and PC2 shifts toward their respective PC1 domain are indicated by curved dashed black arrows. Note the colored β-hairpin loops illustrating their conformational changes (red asterisks). **e** Surface representation of tripartite complex at the level of pore domain of MexB. The three MP domains of MexA-I are contoured. The red arrow indicates the MP domain in contact with the portion of T protomer undergoing the large rmsd corresponding to the "green" segments shown in **d**. MexA-I and MexA-II models are shown in light-green and cyan, respectively. **f** View of the exit gate delineated by the helix gate E121-R124 on the left and the β strand on the right in protomer C of MexB-solo (red) and in protomer O of MexB-trio (cyan). The helix in protomer O of MexB-trio moves away forming a passage for drug exit that is hampered in protomer C of MexB-solo as shown with the tunnel (blue) calculated using CAVER.

## Discussion

RND antiporters transport substrates and protons in opposite directions through a remote coupling between the TM-domain (proton translocation) and the pore domain (substrate export). The existence of structural and energetic coupling between pore domain and proton translocation has been suggested[19]. Although interprotomer interactions are necessary[14], we show that they are not sufficient for such proton/substrate coupling. Indeed, proton transport of MexB-solo was weak in the presence or in the absence of substrate (Fig. 1b) suggesting that substrate binding is not sufficient to elicit a closed to open conformational change (Fig. 5a). By contrast, proton transport measurement of MexB-trio (Fig. 1b) indicated a basal activity in the absence of the substrate and a strongly enhanced one in the presence of

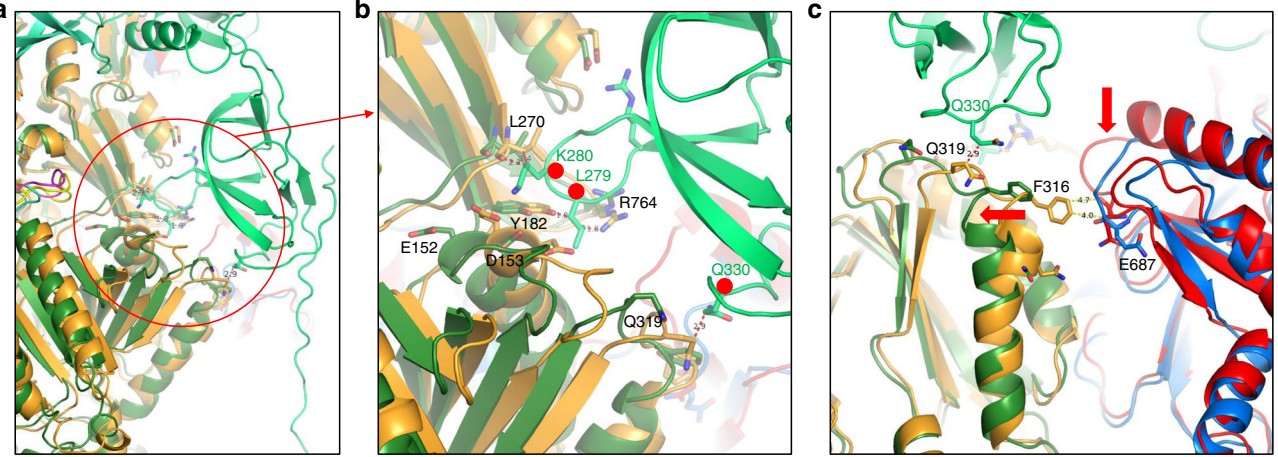

**Fig. 3 Description of the interface between the MexA-II Membrane Proximal domain with the PN2 domain of the protomer T of MexB. a** Superimposition of MexB-trio (magenta, green, blue) and MexB-solo (yellow, orange, red) focusing on the interface between MexA-II membrane proximal domain and MexB protomer T. **b** Enlargement of the region circled in **a**. The red dots indicate MexA zones that would create steric hindrance with MexB-solo inducing protomer T movement to its position in MexB-trio. **c** Same representation showing the interface between protomer T and protomer C/O. The right-to-left shift of PN2 domain is shown by a horizontal red arrow. PC2 of protomer C has lost its stabilization by PN2 of protomer T (vertical red arrow) resulting in a cascading movement that spread up to the PN1 central helix and helix gate.

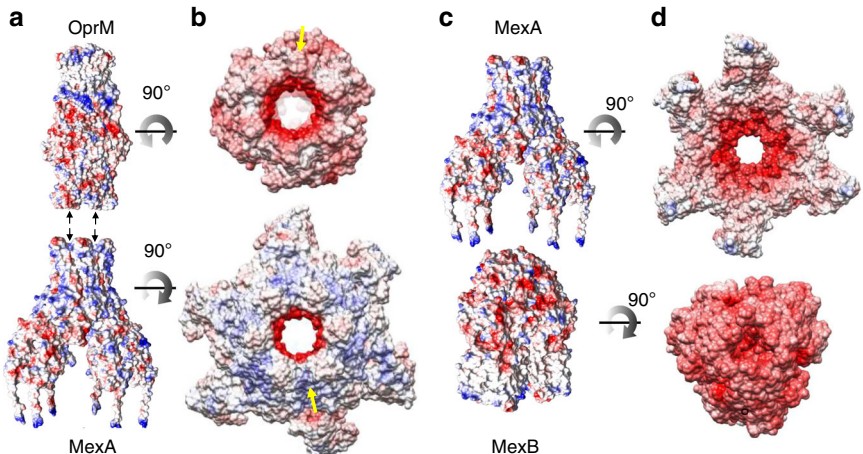

**Fig. 4 Coulomb and electrostatic potential surfaces at the OprM-MexA and MexA-MexB interfaces. a** Side views of OprM and MexA Coulomb potential surfaces. **b** Electrostatic potential surfaces depicting contacts between OprM and MexA (90° rotation with respect to orientation in **a**) with a strong complementarity of potential surfaces, negative charges of OprM facing the positive charges of MexA and vice versa as illustrated with yellow arrows. This surface complementarity is indicative of strong energy binding of tip to tip interaction between OprM and MexA. **c** Side views of MexA and MexB Coulomb potential surfaces. **d** Electrostatic potential surfaces depicting contacts between MexB and MexA (90° rotation with respect to orientation in **c**). No complementary potential is observed indicative of reduced energy binding which is in favor of alternate conformational changes for drug extrusion cycle. Electrostatic potential surfaces were generated by using Amber ff14SB force field ($\varepsilon_{in} = 4$).

substrate. Hence, in the presence of OprM, the hexameric MexA arrangement depicted in the tripartite complex imposes steric constraints on the LTC conformations, notably between MexA-II MP domain and MexB protomer T. Induced conformational movements spread up to the adjacent protomer C causing the transition from LTC to LTO with the exit gate opening (Fig. 5b). MexA binding on MexB elicits the concerted C to O conformational change with the O state representing a higher energy conformation[44]. Our interpretation is that the efflux pump is under the dependence of an allosteric conformational regulation mediated by MexA which, upon binding, restricts the conformational ensemble of MexB regardless of the presence of substrate. If present, the substrate would affect the probabilities in favor of these transitions as proposed for MFS exporters[45] (Fig. 5c).

Molecular dynamics (MD) simulations have provided a mechanistic understanding of the translocation of several substrates based on specific substrate recognition sites i.e. access pocket, binding pocket and exit gate[15,16,46,47]. The recent study of the conformational changes of AcrB-solo, calculated by a hybrid coarse-grained force field in the presence of an indole in the T monomer, is in favour of our hypothesis[48]. In their dynamic calculation of AcrB monitoring the changes of the exit channel, a measurement of the center of mass distances between Q124 and Y758 in close proximity of our chosen reporter pair of amino acids (E121-S756 on Fig. 2d) indicated that the exit gate is closed when no indole is present in the distal pocket. MD simulations of doxorubicin and indole transport through the exit gate occurred over a smooth free energy profile when the exit gate is open[46,48]. On the contrary, a clear peak force of ~8 kcal/(mol Å) is required

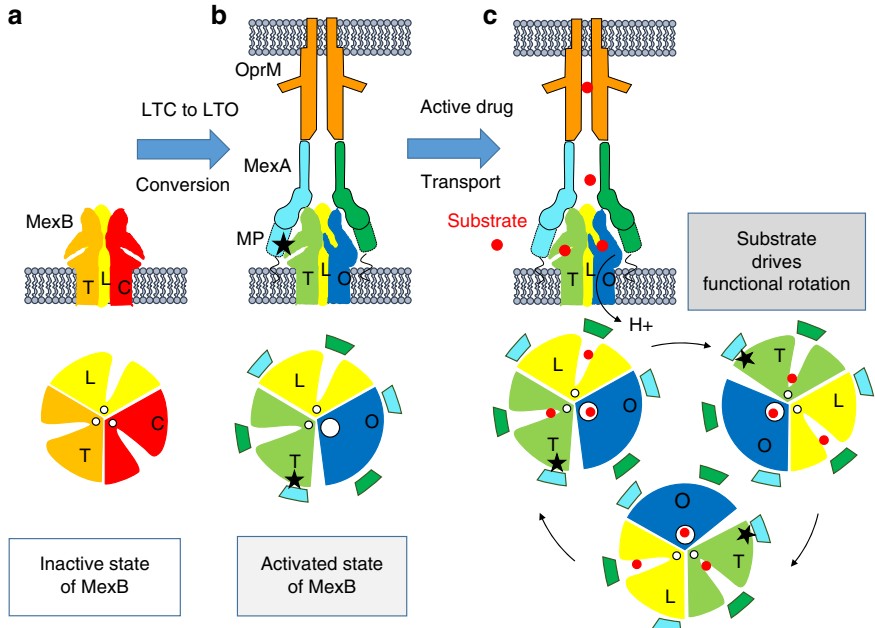

**Fig. 5 Schematic representation of the conversion of MexB from an inactive to an active conformation.** Side view representations of MexB trimer, MexA (green, cyan,), OprM (orange), and lipid membrane (gray) and sectional view representations of MexB and MexA-MexB interaction at the level of the pore domain. The grooves indicate the substrate entry pathway from the periplasmic space and the white disk the exit gate for drug release in the funnel toward OprM. **a** MexB in lipid membrane is inactive. Its conformation states Loose (L), Tight (T) and Closed (C) are colored yellow, orange and red respectively. All exit gates are closed (small white disk). **b** The hexameric MexA arrangement imposes steric constraints on the LTC conformations. Interaction of membrane proximal domain (MP) MexA-II (cyan) with protomer T (black star) induces conformational movements spread up to the adjacent protomer C causing LTC to LTO conversion with the opening of the exit gate (large white circle). MexB is in an activated state. **c** The presence of substrate will favor the transition for the sequential functional rotation mechanism. Activated state of MexB being sustained by its interaction with the three MexA-II MP domains (black stars).

when the exit gate is closed, suggesting that the drug transport is unlikely when the exit gate is closed[48]. This result strongly supports our experimental data on MexB-solo where we observed a closed exit gate for the three protomers and the importance of the exit gate. Moreover, starting from the asymmetric form LTO, MD simulation indicated that in the absence of indole in the protomer O and L, the monomer T remains in the L state and disrupts the functional rotation suggesting that in the absence of substrate such a starting asymmetric form may not exist since it is not regenerated. So another mechanism would be required for inducing this starting asymmetric form. According to our functional data, the two protein partners, MexA and OprM, together with meropenem endow a fully active MexB (Fig. 1b purple curve). When one of the actors is missing (OprM (light orange curve), substrate (red curve) in Fig. 1b), the system is able to work but at a very low yield probably due to a loss of the asymmetric form. Our observation demonstrates that the tripartite assembly maintains/regenerates the asymmetric form when the tripartite pump is assembled, regardless of the presence of substrate.

It has long been believed that large conformational changes mediated by the α-helical coiled-coil hairpins and flexibility of the hinge between the α-helical hairpin and lipoyl domain in the MFP would likely affect the assembly and overall dynamic of the pump[49,50]. Recently, an atomistic model of MexAB-OprM based on molecular dynamics simulations coupled to covariance analyses provided additional insights into the conformational flexibility of MFPs as well as on the transition states at the MexA-OprM interface[43]. However, the latter focused on the possible allosteric effect of MexB, mediated by MexA, on the opening of OprM. We emphasize here that a reciprocal effect exists (OprM-induced activation of MexB via MexA) and that a mutual interplay exists between the respective partners of the pump. Together

cryo-EM structures and experimental in vitro evidence show that MexB sequentially switches from an inactive locked conformation (Fig. 1b, light and dark green curves), to a suboptimal active state where MexA associates with MexB (Fig. 1b, light orange curve), and finally to a fully active state in the tripartite complex (Fig. 1b, purple curve). The mere recruitment of MexA by MexB is not sufficient to elicit full activity of the pump, probably because OprM is needed for the correct positioning of MexA domains. In this sense, the structure of the bipartite RND CusBA complex formed in the absence of the third partner CusC shows that the equivalent of MexA-I (CusB-I) interacts with CusA in a similar fashion as MexA-I does with MexB[51]. But the positioning of CusB-II on CusA is different from that of MexA-II on MexB and more importantly CusB trimer is symmetric indicating that three partners are required to get a full activity of the RND transporter. Therefore, to achieve this perfect positioning, each MexA molecule seems to have a specialized role in the tripartite assembly as previously proposed for TriABC-OpmH system with a stabilizing role for TriA and a transport stimulating role for TriB[52]. Accordingly, MexA-I trimer could be considered as an anchor to stabilize MexA on MexB while the trimeric organization of MexA-II provides a rigid scaffold for chaperoning MexB to adopt functional LTO conformations for drug export. This conclusion is in line with the experimental evidences that MexA association with MexB is dependent upon the presence of OprM[53] and that suppressor mutations within AcrA restoring the function of an inactivated TolC are located within the AcrA β-barrel domain interacting with AcrB[54]. In other words, the two partners of MexB have a more elaborate role than the mere channel formation through the periplasm and the outer-membrane.

From a bioenergetic point of view, it makes much sense for MexB not to be active constitutively because, in such a case there

would be permanent consumption of protons because potential substrates are likely present anytime in the periplasm and that, in the absence of exit duct, the "transported" molecules would be invariably taken up by MexB.

The vertical corseting effect on MexB by MexA via OprM gears MexB for efficient transport only when the complex is formed assuring the expulsion of the drug out of the cell. The relation between structure and activity transport presented here is likely transposable to homologous assemblies in other pathogenic species (*Salmonella*, *Neisseria*, *E. coli*), the three Gram-negative bacteria belonging to the ESKAPE group[55]. To face the drug efflux in bacterial infection, the RND transporter has been mainly targeted by the development of substrate competitors. Perhaps targeting the inactive form before the tripartite assembly would increase the chance of success. Our identification of the key contacts provides new insights into how molecules might be crafted to selectively bind to the inactive form of RND transporters.

## Methods

**Materials and reagents**. 1-palmitoyl-2-oleoyl-*sn*-glycero-3-phosphocholine (POPC) and 1,2-dioleoyl-*sn*-glycero-3-phosphocholine (DOPC) were purchased from Avanti Polar Lipids (USA). Sodium cholate hydrate, octyl-β-D-glucopyranoside (OG) and *n*-Dodecyl β-D-maltoside (DDM) and graphene oxide suspension were purchased from Sigma-Aldrich. SM2 Bio-Beads were obtained from Bio-Rad. Superose 6 3.2/300 column was purchased from GE Healthcare. C-flat grids were obtained from Protochips.

**Protein preparation**. Two membrane scaffold proteins, MSP1D1 and MSP1E3D1 (genetic constructs available from AddGene) expressed and purified from bacteria, were used to make OprM and MexB nanodiscs respectively. MexB, MexA and OprM membrane proteins were expressed and purified from bacteria as previously described[56]. After purification, protein buffers contained 0.03% *n*-DDM for MexB and 0.05% DDM for MexA and OprM.

**Preparation of OprM proteoliposomes**. DOPC lipids in chloroform are dried under vacuum using a rotatory evaporator (Laborota 4000, Heidolph). The lipidic film is then further dried overnight at room temperature under pallet pump, and resuspended 10 min at 37 °C in 25 mM Hepes pH 7.2, 100 mM $K_2SO_4$, 2 mM $MgSO_4$ to a final concentration of 10 mg mL$^{-1}$ of lipids. The suspension is then sonicated for 10 min (30 s pulse, 30 s pause) at 40 Watts. The liposomes are extruded through 400 nm membranes and 200 nm membranes. β-OG is added to reach a 3.4 detergent/lipid ratio (w/w), during 1 h at 4 °C. Proteins are then added to 1 mL solubilized liposomes at ratio lipids/OprM = 20 (w/w), during 1 h at 4 °C. The detergent is then removed using Bio-Beads at a ratio Bio-Beads/detergent of 30 (w/w), during 3 h at 4 °C.

**Preparation of MexAB proteoliposomes**. DOPC lipids in chloroform are mixed with 3 mg cholesterol. The suspension is then dried under vacuum using a rotatory evaporator (Laborota 4000, Heidolph) and then further dried overnight at room temperature under pallet pump. The following day, lipids are resuspended 10 min at 37 °C in 25 mM Hepes pH 7.2, 100 mM $K_2SO_4$, 2 mM $MgSO_4$ to a final concentration of 10 mg mL$^{-1}$ of lipids. The suspension is then sonicated for 10 min (30 s pulse, 30 s pause) at 40 Watts. The liposomes are extruded through 200 nm membranes and 100 nm membranes. Tx-100 is added to reach a 0.9 detergent/lipid ratio (w/w), during 1 h at 4 °C. Proteins are then added to 1 mL solubilized liposomes at ratio lipids/MexB = 20 (w/w) and MexB/MexA = 2.5 (w/w), during 1 h at 4 °C. The detergent is then removed using Bio-Beads at a ratio Bio-Beads/detergent of 30 (w/w), during 3 h at 4 °C. Pyranine was present during vesicle reconstitution and non-incorporated probe was removed by use of PD-10 desalting columns. Vesicle size was checked by dynamic light scattering (Malvern Nanosizer) and Nanoparticle track analysis (NTA, Nanosight NS300 from Malvern Panalytical) and mean diameter was measured to be 110 nm (Supplementary Fig. 10). Homogeneity of the suspension was controlled by negative staining electron microscopy. Sample was deposited on a glow-discharged carbon-coated copper 300 mesh grids and stained with 2% uranyl acetate (wt/vol) solution. Images were acquired on a Tecnai F20 electron microscope (ThermoFisher Scientific) operated at 200 kV using an Eagle 4k_4k camera (ThermoFisher Scientific) (Supplementary Fig. 10). In order to approximate the amount of proteins incorporated into liposomes, the amount of lipids has been estimated by quantitatively determining lipid phosphorus with a linear range after the phosphorus is converted to inorganic phosphate by means of a perchlorate digestion. The stable complex was read with a spectrophotometer at a wavelength of 797 nm. Based on a standard curve, a quantity of 29 nmol of lipids has been determined for 15 μL of proteoliposomes (1). Assuming an average proteoliposome diameter of 110 nm according DLS

measurement and a surface area per lipid molecule to be 0.70 nm², the average number of lipid molecules per proteoliposome is estimated to be 99,185 (2). From (1) and (2), there were ~1.76 10$^{11}$ liposomes per 15 μL of solution (3). Then the amount of proteins has been estimated on SDS-PAGE gel, using the Image Lab software from a range of purified protein. A 15 μL of proteoliposomes contained 271 ± 6 ng of MexA and 165 ± 20 ng of MexB corresponding to 40.7 10$^{11}$ MexA monomer and 8.79 10$^{11}$ MexB monomer (4). From (3) and (4) we estimated that our reconstitution procedure gives rise to 23 MexA and 5 MexB monomers (i.e 1.6 MexB trimers) per a 110 nm liposome.

**Stopped flow measurements**. Stopped flow fluorescence experiments were performed using a Biologic SFM 3000 instrument equipped with a short pathlength optical cell (1.5 mm, "FC15" cell). To detect emitted pyranine fluorescence, we used excitation wavelengths of 455 nm, and wide band emission filters centered in the region of pyranine fluorescence (MTO DA 531, obtained from MTO, Massy (France)). As schematically represented in Fig. 1a, proteoliposomes, loaded in syringe 1, were mixed with syringe 2 filled with an acidic buffer (Hepes 20 mM, $K_2SO_4$ 100 mM, $MgSO_4$ 2 mM) at a 13:3 vol/vol ratio, leading to an overall ΔpH of 1 pH unit. Measurements were performed at 20 °C. Linearity of pyranine fluorescence over the pH range has been assessed (Supplementary Fig. 11). Vesicle concentration was 1 mg/ml and meropenem concentration was 5 μM.

**Formation and purification of tripartite MexAB-OprM complex**. POPC lipids were dissolved in chloroform, dried onto a glass tube under steady flow of nitrogen and followed by exposure to vacuum for 1 h. The lipid film was suspended in the reconstitution buffer (100 mM NaCl, 10 mM Tris/Cl, pH 7.4) and subjected to 5 rounds of sonication (sonicator tip, 30 s pulse, 30 s pause in a glass tube) at 4 Watts. Lipid concentration was quantified by phosphate analysis[57].

MexAB-OprM complex was assembled according to the protocol previously described[38] with slight modifications. Briefly, insertion of OprM in nanodisc (OprM-ND) was performed as follows. OprM solution was mixed with POPC liposomes and MSP solution at a final 36:1:0.4 lipid/MSP/protein molar ratio in a 10 mM Tris/Cl, pH 7.4, 100 mM NaCl and 15 mM Na-cholate solution. Detergent was removed by the addition of SM2 Bio-Beads into the mixture shaken overnight at 4 °C. A mixture of MexA, MexB, MSP proteins and POPC liposomes was prepared at a final 32.6:1:0.5:4.5 lipids/MSP/MexB/MexA molar ratio in a 10 mM Tris/Cl, pH 7.4, 100 mM NaCl and 15 mM Na-cholate solution. Detergents are removed by the addition of SM2 Bio-Beads shaken overnight at 4 °C. The addition of OprM-ND to this sample led to the formation of tripartite complexes. To enrich the tripartite complexes from the initial mixed samples, the MexAB-OprM containing sample was subjected to SEC using a Superose 6 column pre-equilibrated with detergent-free buffer, which was also used as running buffer. The fraction containing the complexes was then directly used for cryo-sample preparation.

**Cryo-EM specimen preparation and data collection**. Holey carbon film 200-mesh R2/1 C-Flat grids (Protochips) were first glow discharged on the carbon film side before depositing 3 μl of 0.5 mg ml$^{-1}$ suspended graphene oxide sheets (Sigma-Aldrich) for 60 s before blotting and air drying over several hours. The grid was UV treated before depositing 3 μl of protein solution at ~0.5 mg ml$^{-1}$ for 60 s at the opposite of the carbon film side. The grid was blotted with ash-less filter paper (Whatman) and plunge frozen in liquid ethane using EMGP (LEICA) under controlled temperature, hygrometry respectively at 4 °C and 80% humidity. The grids were transferred and stored in liquid nitrogen before imaging. Electron microscopy images were recorded on a FEI Titan Krios electron microscope at 300 keV equipped with a post-column GIF Quantum energy filter (20 eV) (Gatan). Dose fractionated images were recorded on a K2 Summit direct detector (GATAN) in a counting mode with a pixel size of 1.36 Å. A data set of 3358 micrographs was collected over a 72 h session. Each micrograph was collected as 40 movie frames for 7 s with a dose rate of 6 e Å$^{-2}$ s$^{-1}$. The total dose was about 42 e Å$^{-2}$. Images were recorded using the automated acquisition program EPU (FEI) with defocus values ranging from −1 to −2.4 μm, a spot size of 6, a condenser aperture of 50 μm, and an objective aperture of 100 μm.

**Cryo-EM data processing**. All movie frames were corrected for gain reference, and aligned using Motioncorr2[58]. Contrast transfer function (CTF) parameters were estimated using Gctf[59]. Further image processing was started in RELION 2.1 and finalized in RELION 3 from the polish step. Initial particle picking of the complex consists in a manual-picking of 1181 particles to calculate 2D references. These 2D templates were low-pass filtered to 20 Å and used for automated picking of all micrographs. A total of 364,914 particles from 3068 micrographs was picked, binned by 4 and extracted with a box of 128 pixels. Several 2D classifications were performed to remove defective particles and led to a subset 96,755 particles. An initial model was calculated from a particle subset of the 2D classes and low-pass-filtering to 60 Å limiting reference bias during 3D classification.

The 3D classification showed similar classes except for one which was removed. The new particle set comprising 90,853 particles was submitted to a 3D auto-refinement process with no symmetry imposed and yielded a 6.9 Å resolution reconstruction for particles binned by 2. The set of unbinned particles was then

submitted to particle polishing (correcting beam-induced motion and radiation-damage weighting) in RELION 3. The 3D refinement and post-processing steps comprising 82,432 particles yielded a 3.1 Å resolution reconstruction of the OprM-MexAB complex with C1 symmetry imposed. A visual inspection of the complex showed a symmetric organization of MexB trimer and an apparent six-fold symmetry of OprM suggesting misalignment and structural heterogeneity.

To correct for misalignment, a masked 3D classification without alignment was performed on the particle set by focusing on MexB. A mask on MexB part was generated from an erased map using the volume erase function in Chimera[60]. The particles were sorted out in three classes representing 30.6%, 40.5% and 28.9% of the particles. The three classes were processed separately using 3D refinement with no symmetry imposed and led to 3D reconstructions at 3.30, 3.25, and 3.41 Å resolution, respectively. One class differed to the other by a 120° rotation of MexB trimer exhibiting asymmetric conformation. The misalignment of the particles was overcome by aligning the reconstructions and merging the particles and calculating a merge volume using 3D refinement.

To sort out particle heterogeneity on OprM, a mask was calculated on the OprM-ND volume generated from the reference map using the volume erase function in Chimera. The particles were sorted out in two main classes representing 43.5% and 52.2% of the particles, two 3D reconstructions were calculated using 3D refinement with no symmetry imposed at 3.21 and 3.15 Å resolution, respectively. The self-assembly process leads to two complexes geometrically related by a 60° rotation of MexB. The hexameric organization of MexA coupled OprM trimer to MexB trimer leading to an uncoupling position/symmetry of OprM from MexB (Supplementary Fig. 2).

For MexB-ND, a procedure similar to that used for tripartite complex has been applied. First a manual-picking of 329 particles was used to calculate 2D references for automated picking. A total of 212,667 particles from 3068 micrographs. Several 2D classifications were performed to remove defective particles and particles corresponding to tripartite complex and led to a subset of 42,317 particles. An initial model was calculated from this particle subset. The 3D classification showed three similar classes except for one that was removed. The new particle set comprising 41,604 particles submitted to 3D refinement yielded a 5.6 Å resolution reconstruction with no symmetry imposed. The set of unbinned particles was then submitted to particle polishing in RELION 3. After 3D refinement and post-processing steps with C1 symmetry imposed, the resolution of MexB-ND reconstruction was improved to 4.5 Å (Supplementary Fig. 5).

The resolution estimation was calculated onto two separately 3D refined half-reconstructions with the Fourier shell correlation criterion at 0.143. Local resolutions were calculated using ResMap[61] and EM density maps corrected for the modulation transfer function of the detector and sharpened by applying a negative B factor determined by the post-processing function of RELION were visualized using UCSF Chimera.

**Model building and structure validation.** The complex assembly structure was solved first. The three proteins were fitted independently in the cryo-EM density using Chimera. For MexB the X-ray structure 3W9J from the Protein Data Bank (PDB: http://www.rcsb.org) was used for the fitting then the [252]-KVNQD-[256] β-hairpin, corresponding to the only obvious modification, was manually modified in Coot[62]. For MexA, the molecule B from the X-ray structure 2V4D was used for the fitting. As the orientation of the respective domains revealed to be very different in the X-ray structure and in the assembly complex, the structure was separated in four domains that have been fitted independently using as a guide the structure of AcrA from the *E. coli* pump (5NG5)[28] followed by a manual correction of the linkers. MexA protomer II was positioned using duplication of MexA protomer I. The membrane proximal domain orientation was corrected by an individual domain fitting. Empty extra densities were present at both the N- and C-terminus of MexA for both monomers. Then it has been possible to build the 12 missing residues on the N-terminus, connecting MexA to the nanodisc artificial membrane. The C-terminus side has been extended of six residues but still fifteen are missing in the final structure. The resulting dimer of MexA was triplicated to complete the hexamer. For OprM, the X-ray structure 3D5K[63] was positioned in one of the two possible orientations and the periplasmic helices were opened manually in Coot. The second possible position of OprM has been refined in a second step. The resulting assemblies were refined in Phenix[64,65] by a rigid body refinement followed by simulated annealing and energy minimization rounds in real-space without applying NCS (Non-Crystallographic Symmetry) constraints. Manual corrections were performed in Coot after each refinement cycle. The final model was submitted to MolProbity program[66] for geometric validation. For the MexB structure resolution, the refined MexB structure from the complex was used to start with. The same procedure was used, first using Chimera for initial fitting, followed by an alternating use of Coot and Phenix for rebuilding and refinement respectively. Data collection, refinement and validation statistics are represented in Supplementary Table 3.

The calculation of the internal tunnels was performed using CAVER[67] using the following settings: minimum probe radius of 1.2 Å, shell depth of 4 Å, shell radius of 6 Å, and the starting point coordinate at the center of the distal-binding pocket (middle of the segment between Cα of residues Q125 and F610).

**Interface analysis.** The interfaces (OprM-MexA and MexA-MexB) were characterized using three quantities: the gap volume, the change of accessible surface area upon binding (ΔASA), and the gap index[46,47]. The gap volume was calculated using the procedure developed by Laskowski[68], which estimates the volume enclosed by any two molecules. The change of solvent accessible surface area on complexation (ΔASA) is defined as:

$$\Delta ASA = \frac{ASA_A + ASA_B - ASA_{AB}}{2} \qquad (1)$$

where A and B represent the monomeric states, AB is the dimeric state. The solvent accessible surface area was calculated by using the same radii as the gap volume calculation[68]. The gap index evaluates the complementarity of the interacting surfaces in a protein-protein complex and is defined as Gap index = Gap volume/ ΔASA. Typical values of the gap index range from $1 \rightarrow 5$. Lower values characterize interfaces with better structural complementarity. In general, protein homodimers have significantly smaller values than heterodimers[69].

The Poisson-Boltzmann (PB) equation model treats solvent as a continuum medium with high dielectric constant. Biomolecules are considered as low dielectric cavities made of charged atoms. Ions in the water phase are modeled as non-interacting point charges and their distribution obeys the Boltzmann law. The resolution of the PB equation allows us to determine the electrostatic potential around the proteins under investigation[70,71].

In our calculation, Poisson-Boltzmann equation was solved using DelPhi 8.0[72]. The protein is assumed to be a homogeneous low dielectric medium $\varepsilon_{in} = 2, 4$, and 10, whereas the solvent is modeled by a high dielectric constant ($\varepsilon_{out} = 80$ at 298 °K for water). We assigned the charges by using Amber ff14SB force field[73].

In order to have an estimation of the binding energy, we expressed the binding free energy of the two proteins in the solvent as:

$$\Delta G_{binding} = G_{AB} - (G_A + G_B) \qquad (2)$$

where $G_{AB}$ is the total free energy of the protein-protein complex in solvent, $G_A$ and $G_B$ are the total free energy of the unbound proteins A and B in solvent. Furthermore, the free energy of each individual entity, $G_i$, can be written as:

$$G_i = E_{MM} + G_{solvation} - TS \qquad (3)$$

where $E_{MM}$ is the average molecular mechanics potential energy, $G_{solvation}$ is the free energy of solvation. $TS$ refers to entropic contribution to the free energy where $T$ and $S$ are the temperature and the entropy, respectively.

The average molecular mechanical potential energy can be written as the sum of bonded and non-bonded terms, as following:

$$E_{MM} = E_{bonded} + E_{nonbonded} = E_{bonded} + E_{vdW} + E_{el} \qquad (4)$$

where $E_{bonded}$ represents the bonded interactions that include the bond, angle, dihedral, and improper dihedral interactions. $E_{nonbonded}$ is the non-bonded interactions including both van der Waals ($E_{vdW}$) and electrostatics ($E_{el}$) interactions, which are modelled using a Lennard-Jones and Coulomb potential function, respectively. Considering that for the partners the same structures in the complex, the change in bonded interactions is equal to zero.

The solvation free energy can be defined as the amount of energy associated with dissolving a solute in a solvent as following:

$$\Delta G_{solvation} = \Delta G_{polar} + \Delta G_{np} \qquad (5)$$

where $\Delta G_{polar}$ is the polar solvation energy, which can be estimated by solving the PB equation. The latter, $\Delta G_{np}$, is the non-polar solvation energy component term and it is usually calculated via linear formula with respect to Δ ASA of the protein and protein complexes:

$$\Delta G_{np} = -2a\Delta ASA - b \qquad (6)$$

where ΔASA is the buried solvent assessable surface area (see Eq. (1)), a and b are two energetic parameters. Common values are $a = 0.00542$ kcal mol$^{-1}$ Å$^{-2}$ and $b = 0.920$ kcal mol$^{-1}$.

**Reporting summary.** Further information on research design is available in the Nature Research Reporting Summary linked to this article.

## Data availability

Data supporting the findings of this paper are available from the corresponding authors upon reasonable request. A reporting summary for this Article is available as a Supplementary Information file. The cryo-EM maps and the atomic models have been deposited in the Protein Data Bank and EMDB under accession numbers 6TA6 and EMD-10395 for the MexA–MexB-trio section, 6TA5 and EMD-10372 for the MexA-OprM section and 6T7S and EMD-10371 for the MexB-solo (see Supplementary Table 1). All other data that support the findings are available from the corresponding authors upon reasonable request. Source data are provided with this paper.

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

## Acknowledgements

This work was supported by French national research agency ANR (Mistec ANR-17CE11-0028, ANR-16-CE11-0001-01), the FRM program (FRMDBF20160635738), Bordeaux INP and Conseil Régional de la Nouvelle Aquitaine. This work has benefited from the facilities of UMS3033/US001-IECB, and the platforms of the Grenoble Instruct-ERIC Center (ISBG: UMS 3518 CNRS-CEA-UGA-EMBL) with support from FRISBI (ANR-10 INSB-05-02) and GRAL (ANR-10-LABX-49-01) within the Grenoble Partnership for Structural Biology (PSB). We thank Nathalie Mignet (Chemical and Biological Technologies for Health Laboratory, Paris Descartes University) for the DLS measurements. We acknowledge the European Synchrotron Radiation Facility for provision of beam time on CM01. We would like to thank Grégory Effantin (ESRF) and Armel Bezault (IECB) for assistance. Q.C. is financed by ANR-16-CE11-0001-01. D.S. and D.P. are recipient of a MENRT fellowship. M.G. is recipient of a fellowship from Bordeaux-INP and Conseil Régional de la Nouvelle Aquitaine.

## Author contributions

M.G., D.S., M.D., L.D. prepared MexAB-OprM sample. I.B., G.P., C.G. developed protein purification protocols. M.G., G.S., J-C.T., O.L. carried out cryo-EM imaging and single-particle analysis. I.B., G.P. built structural models and generated tunnel simulation. E.F. produced energetic calculation. D.P., Q.C., M.P. performed activity transport. M.G., D.P., D.S., E.F., M.P., L.D., I.B., and O.L. interpret the data and wrote the paper with the contribution of all the authors. M.P., L.D., I.B., and O.L. designed the research.

## Competing interests

The authors declare no competing interests.
