## [Peer Review File · Nature Communications]

Reviewers' comments:

Reviewer #1 (Remarks to the Author):

Glavier and colleagues report on their studies of the tripartite multidrug efflux pump from *Pseudomonas aeruginosa* consisting of MexB, a membrane fusion protein MexA, and an outer membrane factor OprM. The authors measure proton ion transfer in a reconstituted system, noting that all three components plus the substrate are required to activate transport, as indicated by an increased acidification in their proteoliposome assay. They have also solved by cryo-EM the structure of MexB alone (MexB solo) and MexB with MexA and OprM (MexB trio) to stated resolutions of 4.6 Å and 3.2 Å, respectively. The authors noted previous work on deciphering the mechanism of substrate transport in tripartite systems, with the current models postulating a variety of allosteric states for *E. coli* AcrB ("functional rotation mechanism") (Pos, 2009 (ref 16)) and a substrate shuffling mechanism within a large cavity with multiple binding sites ("multi-drug oscillation hypothesis") (Yamaguchi et al, 2015 (ref 20)). Glavier and colleagues have interpreted their EM data to show an "LTO" conformation for trimeric MexB, one in which the MexB protomers adopt asymmetric conformations in the presence of MexA and OprM. By comparing MexB solo and MexB trio models, they propose that the exit tunnels for MexB are closed by the movement of helix E121-R124 when MexB is not associated with partners MexA and OprM. They then examine the surface properties of all three components, noting that the charge interactions at the interface between MexA and MexB are not particularly complementary and they conclude that this could therefore allow for MexB to undergo conformational change during substrate transport.

General Comments

While the work in this paper is interesting, especially with regards to determining the intricacies of how tripartite systems transport substrates, this paper would benefit from addressing methodological issues as well as improving coherence and comprehensibility of presentation. One issue is that there is already a MexA/MexB/OprM structure solved by Tsutumi et al (2019) (ref 19) which has completed substantial work in explaining the functioning of this system. The authors have not addressed any differences between their structures and those solved by Tsutumi et al (2019).

The interpretation of MexB in "LTO" conformation when associated with MexA and OprM, as supposed to other possible conformations, needs further substantiation (Fig 2). The statistics supporting this claim are not strong, especially given the low resolution of the MexB solo. The authors have highlighted in grey parts of the MexB structure where the root mean square deviation between MexB solo and MexB trio is less than 2 Å. Comparison of the MexB solo and MexB trio access tunnels seem to show possible conformational change between the structures, but this is hidden as Supp Fig 7. It should be noted that Tsutumi et al (2019) also found an asymmetric MexB structure in their full MexA/MexB/OprM complex, but concluded that it was not drug transport competent even in this asymmetric state. Furthermore, the claim that MexB solo adopts a different "LTC" conformation is also made without presenting a convincing comparison to both the MexB trio and available MexB crystal structure (Stenphauser et al, 2009 (ref 7)) or MexB from Tsutumi et al (2019), weakening this claim (Supp Fig 3).

The report also would be strengthened with additional experimental evidence and further controls. In the proteoliposome acidification assay, the authors attempt to show that proton and substrate transport are coupled only when MexB interacts with MexA and OprM. There is no proper positive control to show that their system works, or visual confirmation that proteoliposomes are formed as described. Fig 1a is inadequately described to explain how the assay works. The proton transport

incompetent MexBD407N is only tested with MexA + meropenem but not all three components together (MexBD407N + MexA + OprM/ MexBD407N + MexA + OprM + meropenem). Supp Fig 1 would be strengthened if these controls could be included.

The authors do not seem to test the tripartite system with any drugs or drug inhibitors, which limits the contribution to the understanding of the functioning of the system. The paper shows no work into the dynamics of the transporter, and hence cannot adequately discriminate in favour of any of the current models of transport.

Do the authors observe the hexagonal lipid packing in the interior as seen in AcrB ? (Qiu, W., Fu, Z., Xu, G.G., Grassucci, R.A., Zhang, Y., Frank, J., Hendrickson, W.A., and Guo, Y. (2018). Structure and activity of lipid bilayer within a membrane-protein transporter. Proc Natl Acad Sci USA 115, 12985-12990.)

Other technical and minor comments

1) The claimed resolutions of Mex B trio and Mex B solo are overestimated. The FSC curve shows the introduction of spurious correlations, likely the result of the particle being at the edge of the mask in certain orientations. This FSC artefact disappears when the phase randomised mask is used which indicates an actual resolution of 6.25 Å (Supp Fig 2d) or 7 Å (Supp Fig 2g) for MexB trio and 8 Å (Supp Fig 5e) for MexB solo. Both MexB trio and MexB solo maps also appears to have been overfitted, with noticeable appearance of noise in some regions.

2) The authors mention a 12 amino acid tail in their model but it would be better to explain the significance of the extra density.

3) Manuscript referencing needs attention in places. For example, in the first sentence there is no reference for "In Gram negative bacteria, tripartite multidrug efflux systems export biological metabolites and antimicrobial compounds, thereby contributing to bacterial resistance". The fourth sentence in the introduction also does not have a reference.

4) The text is unclear in several places. There are many misleading sentences in the introduction which don't provide an adequate interpretation of the literature cited. For example, "It has been proposed that substrates (drug and proton) would stabilize the asymmetric LTO structure" is not a good description of the cited literature which describes a model in which AcrB can adopt different conformations depending on the substrate concentration. Many other such examples are present.

5) The authors should take care to eliminate typos in their manuscript (e.g "seconde" for "seconds").

6) Abstract, line 4 "an energetically uphill drug pathway"

7) Figure 1, separate arrows for proton and meropenem in cartoon schematic.

8) There is a 10 microsecond burst phase in Figure 1. Are there any events on longer or shorter periods?

9) P 5, line 7 sing->using

10) Supplementary figure 1 "MexA transporter is present under its wild type form..." might be easier as "as wild type..."

11) Supplementary figure 5 "Representative 2D class averages of particles extracted from the micrographs used for analysis of tripartite complexes " – this is in fact only for MexB in discs.

12) Supplementary Table 3 – some typo for initial particles no – it seems odd that these are exactly the same in the two adjacent columns

Reviewer #2 (Remarks to the Author):

In this study, functional and structural approaches are used to define a possible conformational activation of an efflux pump MexAB-OprM from *P.aeruginosa*. The reconstitution studies are consistent with previous reports about MexAB as well as other RND transporters that the pump MexB is stimulated by the MFP protein MexA. The cryo-EM analyses of MexB alone and in the MexAB-OprM complex in nanodiscs suggested that MexB alone adopts an asymmetric state, in which one of the protomer is in a closed inactive conformation. In contrast, MexB assembled into MexAB-OprM complex adopts an asymmetric functional state, similar to that described before for AcrAB-TolC.

The closed inactive conformation was not observed before in these transporters. The finding of this conformation is novel, albeit it requires further validation. However, the bulk of the results is largely confirmatory.

The structure of MexB is resolved at 4.6 Å, which is quite low and raises concerns about possible overinterpretation. This novel closed conformation requires further validation. For example, mutagenesis stabilizing the closed MexB state and functional characterization of the mutants including an investigation how such mutations affect the assembly and stability of the complex are essential. Without such validation, the result is open to a possibility that this conformation is an artifact.

The mechanism of the activation remains unclear. There are numerous *in vivo* and *in vitro* studies demonstrating that an MFP and a transporter form a complex even in the absence of an outer membrane channel. In contrast, the authors claim that OprM is required for MexA to bind MexB. This needs further experiments to demonstrate.

Although a "sequential" switching from a closed to an open conformation is proposed, the sequence of events remains unclear.

Reviewer #3 (Remarks to the Author):

This manuscript provides new structural insight into the tripartite multidrug efflux system MexAB-OprM based on nanodisc-reconstituted components and cryo-EM. By analyzing the membrane-embedded MexB trimer before and after assembling the tripartite complex, the authors identify a key role of MexA and OprM in setting the transporter into a functional state. A new stop flow approach demonstrates the *in vitro* proton transport activity of the fully assembled MexAB-OprM complex.

The topic explored here is of great interest and the claims are certainly novel and potentially impactful. I also appreciate the combination of biochemical data with structural data. However, the design and the key data sets for the *in vitro* activity assay have been published before by the authors (Ref 22, Verchère et al. 2015, Nat Commun 6, 6890). The new assay presented here has a higher time resolution but the data shown do not add new information. In fact it remains unclear if the MexAB-OprM complex is active in POPC used for the preparation of the nanodiscs samples, since proteoliposomes composed of DOPC and DOPC/cholesterol were used in the transport assay. The novel assay would allow testing this issue and thereby supporting the relevance of the presented structures based on POPC nanodiscs. An additional flaw concerns the Method part that lacks at several locations essential information that should be added at least to the supplement to allow reproduction of the data:

- When was pyranine present during the vesicle reconstitution? Was non-incorporated pyranine

removed, how?

- The precise emission wavelength and band width for detection of pyranine are not give; at which temperature was the stopped flow assay performed? What was the vesicle concentration? What was the final meropenem concentration?

- Essential data on the final proteoliposome preparations are lacking (which I could also not find in Verchère et al. 2015, Nat Commun 6, 6890): what was the protein/lipid ratio finally in the vesicles, what was the size of the vesicles, how many protein-free vesicles were present?

A minor point concerns the use of abbreviations in the abstract that are not explained (LTO, LTC, LTO).

In our response below, remarks of the Reviewers are reproduced *in extenso* (in black) and we address their comments and concerns point-by-point (in blue). When necessary, corrections or addendum to the original manuscript have been made: they are reproduced here and inserted in the main text (in red).

Reviewer #1 (Remarks to the Author):

Glavier and colleagues report on their studies of the tripartite multidrug efflux pump from *Pseudomonas aeruginosa* consisting of MexB, a membrane fusion protein MexA, and an outer membrane factor OprM. The authors measure proton ion transfer in a reconstituted system, noting that all three components plus the substrate are required to activate transport, as indicated by an increased acidification in their proteoliposome assay. They have also solved by cryo-EM the structure of MexB alone (MexB solo) and MexB with MexA and OprM (MexB trio) to stated resolutions of 4.6 Å and 3.2 Å, respectively. The authors noted previous work on deciphering the mechanism of substrate transport in tripartite systems, with the current models postulating a variety of allosteric states for *E. coli* AcrB (“functional rotation mechanism”) (Pos, 2009 (ref 16)) and a substrate shuffling mechanism within a large cavity with multiple binding sites (“multi-drug oscillation hypothesis”) (Yamaguchi et al, 2015 (ref 20)). Glavier and colleagues have interpreted their EM data to show an “LTO” conformation for trimeric MexB, one in which the MexB protomers adopt asymmetric conformations in the presence of MexA and OprM. By comparing MexB solo and MexB trio models, they propose that the exit tunnels for MexB are closed by the movement of helix E121-R124 when MexB is not associated with partners MexA and OprM. They then examine the surface properties of all three components, noting that the charge interactions at the interface between MexA and MexB are not particularly complementary and they conclude that this could therefore allow for MexB to undergo conformational change during substrate transport.

General Comments

In his general comment, the reviewer has concerns on several aspects of the manuscripts. We have taken care to provide detailed responses after each paragraph, by adding additional materials and modifying the text accordingly.

While the work in this paper is interesting, especially with regards to determining the intricacies of how tripartite systems transport substrates, this paper would benefit from addressing methodological issues as well as improving coherence and comprehensibility of presentation. One issue is that there is already a MexA/MexB/OprM structure solved by Tsutsumi et al (2019) (ref 19) which has completed substantial work in explaining the functioning of this system. The authors have not addressed any differences between their structures and those solved by Tsutsumi et al (2019).

We agree that a comparison with the recently published cryoEM MexAB-OprM structure (Tsutsumi *et al.*, 2019) was missing. Tsutsumi *et al.* (2019) described the structure of MexAB-OprM stabilized in amphipol using full-length-encoded OprM and MexB proteins. A modified sequence of MexA was used. The first cysteine of MexA was removed preventing the attachment of fatty acids via a posttranslational modification. They have identified two

binding modes of OprM to MexA. Like Tsutsumi, we also identified two binding modes which give rise to two MexAB – OprM structures. Although our protocol is not identical to the one used by Tsutsumi *et al.*, these structural similarities indicate that the self-assembly process of the tripartite system is independent of the method used for mixing the components. We did not focus on this aspect because it was not the main scope of the manuscript.

Unlike Tsutsumi *et al.* we used the full-length genes for encoding the three proteins. MexA bears fatty acids attached to the N-terminal cysteine residue. Our structure of MexAB-OprM stabilized in nanodisc gives access to densities likely corresponding to the first amino acids at the N-terminal part of MexA whose flexibility is reduced by its anchorage in the lipid membrane. Therefore, we have been able to extend the structural description of MexA that was not accessible from the crystal structure of MexA and Tsutsumi's structure.

We have improved the text accordingly by adding the following paragraph:

Firstly, the 3.2 Å resolution tripartite complex structure in nanodisc revealed a 1:2:1 stoichiometry of OprM, MexA, and MexB with a OprM trimer, a MexB trimer (MexB-trio) and a MexA hexamer surrounding MexB and interacting with OprM in an open conformation. The α -hairpin domain of MexA contacts the extremity of the periplasmic helices of OprM while the β -barrel domain and the MP domain interact with the funnel domain and the pore domain of MexB respectively. The overall structure was similar to that of MexAB-OprM stabilized in amphipol (Tsutsumi *et al.*, 2019). Likewise, two complexes geometrically related by a 60° rotation of either OprM or MexB were identified (Supplementary Fig. 2f) suggesting two binding modes of OprM to MexA (Tsutsumi *et al.*, 2019).

The reviewer wrote that Tsutsumi *et al.* (2019) “has completed substantial work in explaining the functioning of this system”. We agree that they added information at the level of the binding pocket. The structure of MexAB-OprM with novobiocin is similar to the corresponding drug-free structure with a difference at the gate loop (G675-F680) in binding (T) protomer. They proposed a mechanism of drug transport which is activated by the presence of drug. The latter opens the gate loop giving access to the binding pocket, switching the protomer from resting to binding state (switch from ARE to ABE).

The structure of the gate loop of MexB-trio is similar to that of the drug-free MexB as shown below with the snapshot superimposed structures of MexB-trio (cyan) drug-free MexB (green) and MexB-ABI-PP complex from Nakashima *et al.* (2013) (yellow).

We modified the text accordingly in the following sentence:

“These conformational states were similar, notably in the pore domain (Supplementary Fig. 3), to those of the crystal and cryoEM structures in the absence of substrate, which is in perfect agreement with the functional rotating mechanism.”

Even though Tsutsumi *et al.* observed, like us, that MexB is asymmetric in their tripartite structure, they focused on the drug transport at the level of the binding pocket involving the protomer B/T responsible for drug binding. They did not address the question we addressed in this report concerning the role of the cognate partners of MexB on its functioning. Our findings are based on functional data unveiling new aspects on the functioning of the efflux pump. The structural data highlight a conformational change of a protomer which is not the B/T one responsible for drug binding but the one responsible for the drug exit, deciphering molecular mechanism for another portion of the drug pathway. And we emphasize that MexB activity is governed by its cognate partners that generate its LTO state, regardless of the presence of substrate.

The interpretation of MexB in “LTO” conformation when associated with MexA and OprM, as supposed to other possible conformations, needs further substantiation (Fig 2). The statistics supporting this claim are not strong, especially given the low resolution of the MexB solo. The authors have highlighted in grey parts of the MexB structure where the root mean square deviation between MexB solo and MexB trio is less than 2 Å. Comparison of the MexB solo and MexB trio access tunnels seem to show possible conformational change between the structures, but this is hidden as Supp Fig 7. It should be noted that Tsutsumi *et al.* (2019) also found an asymmetric MexB structure in their full MexA/MexB/OprM complex, but concluded that it was not drug transport competent even in this asymmetric state. Furthermore, the claim that MexB solo adopts a different “LTC” conformation is also made without presenting a convincing comparison to both the MexB trio and available MexB crystal structure (Stenphauser *et al.*, 2009 (ref 7)) or MexB from Tsutsumi *et al.* (2019), weakening this claim (Supp Fig 3).

Tsutsumi *et al.* (2019), noted that in their tripartite complex MexB protomers adopted Access, Resting and Extrusion forms (Loose, Resting and Open forms according to our nomenclature). And in the presence of substrate, MexB protomers adopted Access, Binding and Extrusion forms (Loose, Tight, Open forms). According to this hypothesis, our MexB-trio would be in a LRO conformation. However, we prefer to keep the general nomenclature LTO as the resting state is not yet fully demonstrated.

Our findings are based on functional and structural data unveiling new aspects on the functioning of the efflux pump. Indeed, we provide functional evidence showing that MexB alone is not functional even in the presence of substrate, meaning that the substrate by itself is not able to activate drug transport. A structural comparison of MexB alone (MexB-solo) with MexB in the tripartite complex (MexB-trio) revealed a conformational difference at the level of the exit gate of the extrusion (O) protomer, not only presented in Supplementary Fig. 7 but also appearing in the main text as Fig 2d. The opening of the exit gate is triggered by MexA-OprM association and occurs in the absence of substrate. Our original findings indicate that the cognate partners of MexB play a major role for its transport function and are

complementary to Tsutsumi *et al.* (2019) adding another piece of the puzzle for understanding of drug efflux mechanism.

We have prepared a new figure (that becomes Supplementary Fig. 8) comparing MexB-solo to crystal structures of MexB without added substrate and the inhibitor-bound MexB-ABI-PP.

Supplementary Fig. 8: Comparison of MexB-solo with two crystal structures of MexB.

Superimposition of atomic models of MexB solo in orange, MexB without added substrate in green (PDB: 2V50) and the inhibitor-bound MexB–ABI-PP in light orange (PDB 3W9J)

a, Top view showing structure difference in the encircled protomer C/O. The helix gate is marked with the red arrow. **b**, Close up view of the helix gate position in a close (red arrow) and open state corresponding to a shift of about 6 Å. **c**, View delineating cavities for substrate pathway (dark blue arrow) passing successively along the gate loop (yellow arrow), the switch loop (light blue arrow) and the helix gate (red arrow).

We have added a precision in the following sentence about the rmsd for the helix gate

“By contrast, in the third protomer of MexB-solo which is expected to be involved in drug release as observed in the O protomer of MexB-trio the most striking change (rmsd 1.43 Å) concerned the central helix plus helix gate (amino acids 99 to 124) (Fig. 2c, d).”

The report also would be strengthened with additional experimental evidence and further controls. In the proteoliposome acidification assay, the authors attempt to show that proton and substrate transport are coupled only when MexB interacts with MexA and OprM. There is no proper positive control to show that their system works, or visual confirmation that proteoliposomes are formed as described. Fig 1a is inadequately described to explain how the assay works. The proton transport incompetent MexBD407N is only tested with MexA + meropenem but not all three components together (MexBD407N + MexA + OprM/MexBD407N + MexA + OprM + meropenem). Supp Fig 1 would be strengthened if these controls could be included.

The activity assay is very challenging and is prone to many artefacts. The reviewer is right: many controls must be performed in order to make sure that the observed fluorescence variations indeed reflect activity of the pump and not, for instance, passive leakage of protons through lipid membranes. The corresponding proofs of concepts and, hopefully, convincing data proving that our transport assay is indeed relevant, reproducible and robust have been published previously in Verchère *et al.* (Nature Communications 2015). All controls are described therein, including the one relevantly suggested by the reviewer (MexBD407N + MexA + OprM in the presence or in the absence of meropenem). It can be found in the supplementary materials (Supp Mat figure 6).

Regarding the positive control to show that the system works as suggested, we would refer to the paper Ntsogo *et al.* (Front Microbiol 2015), where we designed a way to make sure that indeed transport occurs upon interaction of MexB with MexA and OprM. This was investigated through the use of MexAB proteoliposomes enriched with biotinylated lipids. The latter were pulled-down with streptavidine beads. The additional presence of OprM in the pellet, indicative of a formation of a stable tripartite complex, was checked by SDS PAGE. Again, systematic controls (empty liposomes, inactive mutants) were performed.

The authors do not seem to test the tripartite system with any drugs or drug inhibitors, which limits the contribution to the understanding of the functioning of the system. The paper shows no work into the dynamics of the transporter, and hence cannot adequately discriminate in favour of any of the current models of transport.

As stated above, we are perfectly aware that controls are mandatory for our experimental approach to be fully convincing. Subjecting our *in vitro* system to drug inhibitors would have been an option and was in fact considered at the initial stages of our project. However, we

have decided to challenge our protocol with more standard (although more tedious to undertake and at least as convincing) procedures: comparison of proteoliposomes in the presence or in the absence of substrate, in the presence or in the absence of energy, in the presence of a wild type versus in the presence of an inactivated version of the transporter.

The only commercially available inhibitor is PA β N (Lomovskaya, O., M. S. Warren, A. Lee, et al. 2001 *Antimicrobial Agents and Chemotherapy* 45(1): 105–116.) but it is known from the original publication that it does not only inhibit efflux pumps but also permeabilizes the outer membrane of Gram-negative bacteria. Hence we think that this inhibitor is not appropriate because its use would lead to a further leakage of protons, irrespective of the presence of protein.

ABI-PP (or D13-9001, see ref. Nakayama K, Ishida Y, Ohtsuka M, Kawato H, Yoshida K, Yokomizo Y, et al. *Bioorg Med Chem Lett* 2003;13:4201e4.) and MBX 3132 (Sjuts, Hanno, Attilio V. Vargiu, Steven M. Kwasny, et al. 2016 *Proceedings of the National Academy of Sciences*: 201602472.) share more relevant properties as they were described to have little effect in terms of membrane permeabilization and efficiency in the potentiation of the action of antibiotic. Both inhibitors have been shown to bind in the deep-binding pocket, the so-called hydrophobic trap, thereby blocking the T to O transition. Although indeed theoretically relevant controls, we think that these molecules would only provide information redundant with what is already known in the literature.

In addition, Tsutsumi *et al.* (2019) has previously described a MexAB-OprM structure with a substrate showing difference at the level of the binding pockets. No difference at the exit gate is observed. In the ms, we explored the drug transport a step further after the binding pocket corresponding to the exit gate for which no inhibitor has been described hitherto. Instead of using inhibitors we decipher in a step-by-step assembly process the functional activity of MexB which is controlled by its cognate partner.

To answer Reviewer's concern about dynamic, several studies from the literature are in line with the model transport presented in the manuscript. A molecular dynamics (MD) simulation showed an assembled MexAB-OprM complex inserted in a POPC lipid membrane (Lopez et al, 2017). OprM conformation with an open periplasmic aperture contacts the α -hairpin tips of MexA hexamer even though there was no bound drug added in the simulation. This MexA-OprM conformation is similar to the one observed in our cryoEM structure. The OprM opening (so-called tip-to-tip interaction) is achieved without the need of substrate, unlike Wang's hypothesis based on the apo-form (Wang *et al.*, 2017). Less details for the MexA-MexB interface are available preventing a comparison with our CryoEM structure.

MD simulations have provided a mechanistic understanding of the translocation of several substrates based on specific substrate recognition sites i.e. access pocket, binding pocket and exit gate (Schulz *et al*, 2010, Schulz, 2011, Vargui *et al* 2018 Atzori *et al*, 2019). The latter formed by residues Gln124, Gln125, and Tyr758 gives access to the central funnel at the top of the periplasmic domain and is only open in the O protomer (Sennhauser *et al*, 2007; Schulz *et al*, 2010, Schulz *et al.*, 2011). In Vargiu *et al.* (2018), the authors showed that the doxorubicin transport through the exit gate occurred over a smooth free energy profile and was facilitated by a continuous hydration layer.

Additionally, a dynamic analysis of AcrB has been very recently published (Jewel *et al.*, 2020 Jan 30), *i.e.* released after the submission of our ms) and strongly support our experimental data and the importance of the exit gate. They have monitored the changes of the exit channel by measuring the centre of mass distances between Gln124 and Tyr758 in their simulations. The exit gate is closed in a model in which no indole/proton has been added to the system. This result is similar to our observation on MexB-solo where we observed a closed exit gate

for the three protomers. Interestingly they estimated the energy required for the indole exit using a pulling force. The pulling force profile is relatively small when the exit gate is open while a more significant peak force of ~ 8 kcal/(mol.Å) is required when the exit gate is closed, suggesting that the drug transport is unlikely when the exit gate is closed. Moreover, starting from the asymmetric form LTO, they observed that indole is only able to change its host monomer (from T to O in their case) with protonation. Thus, the fully functional rotating mechanism ceases to exist if only one of three monomers host the substrate. It has been found that in the absence of indole in the protomer O and L, the monomer T remains in the L state and disrupts the functional rotation. So these results indicate that in the absence of substrate such a starting asymmetric form may not exist since it is not regenerated. So another mechanism is required for inducing this starting asymmetric form. We believe that our experimental data gives credit for an activation of the asymmetric form upon tripartite assembly that would lead to the maintenance/regeneration of the asymmetric form regardless the presence of substrate.

We have improved the text accordingly by adding in the discussion part the following paragraph

Molecular dynamics (MD) simulations have provided a mechanistic understanding of the translocation of several substrates based on specific substrate recognition sites i.e. access pocket, binding pocket and exit gate (Schulz et al, 2010, Schulz, 2011, Vargiu et al 2018 Atzori et al, 2019). The recent study of the conformational changes of AcrB-solo, calculated by a hybrid coarse-grained force field in the presence of an indole in the T monomer, is in favour of our hypothesis (Jewel, 2020). In their dynamic calculation of AcrB monitoring the changes of the exit channel, a measurement of the center of mass distances between Q124 and Y758 in close proximity of our chosen reporter pair of amino acids (E121-S756 on figure 2d) indicated that the exit gate is closed when no indole is present in the distal pocket. MD simulations of doxorubicin and indole transport through the exit gate occurred over a smooth free energy profile when the exit gate is open (Vargiu et al, 2018; Jewel et al., 2020). On the contrary, a clear peak force of ~ 8 kcal/(mol.Å) is required when the exit gate is closed, suggesting that the drug transport is unlikely when the exit gate is closed (Jewel et al, 2020). This result strongly supports our experimental data on MexB-solo where we observed a closed exit gate for the three protomers and the importance of the exit gate. Moreover, starting from the asymmetric form LTO, MD simulation indicated that in the absence of indole in the protomer O and L, the monomer T remains in the L state and disrupts the functional rotation suggesting that in the absence of substrate such a starting asymmetric form may not exist since it is not regenerated. So another mechanism would be required for inducing this starting asymmetric form. According to our functional data, the two protein partners, MexA and OprM, together with meropenem endow a fully active MexB (Fig 1b purple curve). When one of the actors is missing (OprM (light orange curve), substrate (red curve)), the system is able to work but at a very low yield probably due to a loss of the asymmetric form. Our observation demonstrate that the tripartite assembly maintains/regenerate the asymmetric form when the tripartite pump is assembled, regardless of the presence of substrate.

Do the authors observe the hexagonal lipid packing in the interior as seen in AcrB ? (Qiu, W., Fu, Z., Xu, G.G., Grassucci, R.A., Zhang, Y., Frank, J., Hendrickson, W.A., and Guo, Y. (2018). Structure and activity of lipid bilayer within a membrane-protein transporter. Proc Natl Acad Sci USA 115, 12985-12990.)

We agree that it is an interesting question. Nevertheless, the protocol for MexB purification has been performed with detergent unlike Qiu *et al.* who used styrene maleic acid SMA. With our protocol we did not see any regular lipid packing.

Other technical and minor comments

1) The claimed resolutions of Mex B trio and Mex B solo are overestimated. The FSC curve shows the introduction of spurious correlations, likely the result of the particle being at the edge of the mask in certain orientations. This FSC artefact disappears when the phase randomised mask is used which indicates an actual resolution of 6.25 Å (Supp Fig 2d) or 7 Å (Supp Fig 2g) for MexB trio and 8 Å (Supp Fig 5e) for MexB solo. Both MexB trio and MexB solo maps also appears to have been overfitted, with noticeable appearance of noise in some regions.

First of all, we would like to mention that we have applied the gold standard procedure for 3D reconstruction and FSC calculation described by Chen *et al.*, (2013) and implemented in Relion software including the use of a soft mask with the recommended parameters. According to the authors, "this gold-standard FSC weighting procedure produces no overfitting, as expected since the two halves of the data have never been allowed to interact other than to produce the weighting, whereas the more conventional FSC weighting produces a much more significant degree of overfitting".

We agree with the reviewer that compared with the masked FSC curve, the unmasked FSC curve has an atypical shape with an unexpected bump. Interestingly, the unmasked FSC curve of MexAB-OprM structure determined by Tsutsumi *et al.*, (2019) exhibited a shape similar to our unmasked FSC curve. The fact that these two independent studies led to a similar effect on the unmasked FSC probably relies on the shape the tripartite complex. Indeed, the complex is an elongated thin structure 32 nm in length and about 8 nm in diameter that is built in a reconstruction sphere with a diameter of about 1.5 larger than the length of the molecule (rule imposed for the 3D reconstruction). This is an atypical case for single particle reconstruction method since most of the proteins are more or less globular. For the tripartite complex, the reconstruction sphere contains more noise than for globular proteins. This could have an impact on the shape of the unmasked curves. Unfortunately, the unmasked FSC curve of AcrAB-TolC structure (Wang *et al.*, 2017) is not shown avoiding the comparison with MexAB-OprM complex. An additional argument in favour of the effect of the molecule shape is that we have used the same procedure for MexB solo as for MexAB-OprM, and as shown in Supplement Fig 5, the discrepancy between unmasked and masked FSC curves is not observed likely because MexB-solo is more globular.

And ultimately as a proof of the quality of the cryoEM map in term of resolution, we provide snapshots of cryoEM map and fitted atomic model for MexB-solo and MexB-trio.

Model of MexB-solo CryoEM map and fitted atomic model
A) Side view. B) Close-up of the switch loop (615-620). C) Enlarged view of the funnel domain.

Model of MexB-trio. CryoEM map and fitted atomic model.

A) Side view of MexB-trio and MexA. B) Transversal view. C) Close-up of the helices 121-124 and the switch loop (615-620) D) Close-up of the switch loop (615-620). Side chains of aromatic residues are well resolved in agreement with the 3.2 Å resolution

2) The authors mention a 12-amino-acid tail in their model but it would be better to explain the significance of the extra density.

As explained above, the extra densities correspond to the N terminal amino acid segment not described before. We got access to these densities probably because the flexibility of this regions is strongly reduced by its anchorage in the lipid membrane via the fatty acid linked to the Nter cysteine.

3) Manuscript referencing needs attention in places. For example, in the first sentence there is no reference for “In Gram negative bacteria, tripartite multidrug efflux systems export biological metabolites and antimicrobial compounds, thereby contributing to bacterial

resistance”. The fourth sentence in the introduction also does not have a reference.

We disagree. two references are cited for the first sentence.

For the fourth sentence, we have added the following references:

“Du *et al.*, 2018 and Zwama *et al.*, 2018 »

4) The text is unclear in several places. There are many misleading sentences in the introduction which don't provide an adequate interpretation of the literature cited. For example, “It has been proposed that substrates (drug and proton) would stabilize the asymmetric LTO structure” is not a good description of the cited literature which describes a model in which AcrB can adopt different conformations depending on the substrate concentration. Many other such examples are present.

For the sake of clarity this part in the introduction has been rewritten to improve the description of the cited literature.

Crystal structure determination revealed that MexB as well as the *E coli* homologous AcrB occurred as asymmetric homotrimers for which the three monomer conformations representing consecutive states were designated Loose (or Access), Tight (or Binding) and Open (or Extrusion) (Sennhauser *et al.*, 2009; Murakami *et al.*, 2006; Seeger *et al.*, 2006; Sennhauser *et al.*, 2007). A hydrophobic pocket created in the T monomer was described as a substrate binding pocket based on the structures of AcrB-drug complexes showing the binding of minocyclin, and doxorubicin to this pocket (Murakami *et al.* 2006). From the asymmetric AcrB structures, a model for a directional drug transport based on conformational cycling of the monomers (referred also as functionally rotating mechanism) by utilizing proton binding and dissociation has been proposed with the substrate entry happening in the L monomer, followed by conformational change to the T monomer resulting in the substrate binding to the binding pocket and finally the substrate release arose after T monomer conversion to the O monomer. The cycling event was regenerated after the conversion the O monomer to the L monomer. This model has been supported by additional experimental studies (Seeger *et al.*, 2008; Oswald *et al.*, 2016), molecular dynamics simulations (Schulz *et al.*, 2010; Schulz *et al.*, 2011; Yamane *et al.* 2013), and thermodynamic calculations (Mishima *et al.*, 2015; Matsunaga, 2018). As a result of these studies, specific substrate recognition sites have been identified based on AcrB structure. The access pocket is placed in the vestibule region between PC1 and PC2 subdomains and open in protomers L and T (Nakashima *et al.*, 2011; Eicher *et al.*, 2012). The distal pocket separated from the access pocket by the switch loop is located more closely to the funnel domain and open only in the T protomer (Nakashima *et al.*, 2011; Eicher *et al.*, 2012). The exit gate formed by residues Gln124, Gln125 (part of the later called helix gate corresponding to the N α 2' helix (Murakami *et al.*, 2002)), and Tyr758 gives access to the central funnel at the top of the periplasmic domain and is only open in the O protomer (Sennhauser *et al.*, 2007; Schulz *et al.*, 2010, Schulz *et al.*, 2011).

Intriguingly symmetric trimers have been reported for AcrB (Murakami *et al.* 2002) and for other RND transporters (Su *et al.*, 2017; Long *et al.*, 2010; Bolla *et al.*, 2014) i.e CmeB (Su *et al.*, 2017), CusA (Long *et al.*, 2010), ZneA (Pak, *et al.*, 2013), AdeB (Su *et al.*, 2019). Single molecule FRET studies of CmeAB in proteoliposome have suggested that each protomer experiences independent conformational changes (Su *et al.*, 2017) maintaining some uncertainty on the direction of the monomer conversion (e.g. from L to T, T to O and O to L) and on a common cycling mechanism of RND drug transport.

Recent cryo-Electron Microscopy (cryo-EM) studies of AcrAB-TolC and MexAB-OprM systems stabilized with amphiphilic polymer have shown overall similar architectures of six MFP surrounding a trimer of RND and in a tip-to tip interaction with the OMF. Therein, AcrB and MexB adopt an asymmetric LTO conformation in a subset of particles (Wang et al., 2017; Tsutsumi et al., 2019). Interestingly, in both studies tripartite structures have been determined in the presence and in the absence of substrate describing a transport-state and a resting state respectively. According to Wang et al, (2017), the resting state corresponding to the apo form with TolC in closed-state and AcrB in LLL conformation switched to a transport-state in the presence of puromycin in which the AcrB trimer adopted asymmetric LLT, LTT or LTO conformations and TolC is open. For Tsutsumi et al., (2019), the resting state corresponded to the tripartite structure with an opened OprM and an asymmetric MexB trimer as in crystal LTO conformation except for the gate loop in the T protomer. In the presence of novobiocin, T protomer underwent a slight conformation change. The gate loop was stretched and descended allowing novobiocin binding in the distal binding pocket. From all the above, although it has been postulated that the LLL to LTO transition could be driven upon substrate binding (Su et al, 2019; Pos et al., 2009, Wang et al, 2017).

5) The authors should take care to eliminate typos in their manuscript (e.g “seconde” for “seconds”).

Despite a careful inspection, we did not find this typo.

6) Abstract, line 4 “an energetically uphill drug pathway”

The sentence written in the abstract is “an uphill drug pathway” and not “an energetically uphill drug pathway” as pointed out by the reviewer.

To avoid any confusion, in the abstract we replace
“an uphill drug pathway”

by

“a directional drug pathway”

7) Figure 1, separate arrows for proton and meropenem in cartoon schematic.

Correction has been done accordingly.

8) There is a 10 microsecond burst phase in Figure 1. Are there any events on longer or shorter periods?

The stopped-flow apparatus is designed to allow recordings with very short dead time, down to 3 msec. From our experience as well as from the specifications of the supplier, any event monitored below this value shall not be taken into account. For instance, the 10 microsec burst phase should not be interpreted any further as it may arise from mechanical vibration upon triggering of the rapid mixing, electrical noise, temperature equilibration, ... Although we display fluorescence measurements over one second, we systematically record traces over much longer time scales (typically 2 minutes), in order to make sure that additional events do

not take place. We convinced ourselves that this is the case, as shown with the two examples below (screenshots of classical recordings over 1 second, up, or 120 seconds, down).

Note that on longer periods, the fluorescence signals show a continuous decreasing drift, that we ascribe to the slow passive entry of protons that is rate-limited by the membrane potential across the liposome membrane (see Verchère *et al.*, Nature Communications 2015 Supplementary figures 4 and 5).

9) P 5, line 7 sing->using
Correction done

10) Supplementary figure 1 “MexA transporter is present under its wild type form...” might be easier as “as wild type...”

The text has been modified accordingly

11) Supplementary figure 5 “Representative 2D class averages of particles extracted from the micrographs used for analysis of tripartite complexes “ – this is in fact only for MexB in discs.

We agree with the reviewer. We have modified the sentence accordingly which is now;

“Representative 2D class averages of **MexB-ND** particles extracted from the micrographs used for analysis of tripartite complexes (shown in Supplementary Fig. 2a)

12) Supplementary Table 3 – some typo for initial particles no – it seems odd that these are exactly the same in the two adjacent columns

These two cryoEM maps have been obtained with the same initial set of MexAB-OprM particles, but the number of final particles used for final refinement is different.

Reviewer #2 (Remarks to the Author):

In this study, functional and structural approaches are used to define a possible conformational activation of an efflux pump MexAB-OprM from *P.aeruginosa*. The reconstitution studies are consistent with previous reports about MexAB as well as other RND transporters that the pump MexB is stimulated by the MFP protein MexA. The cryo-EM analyses of MexB alone and in the MexAB-OprM complex in nanodiscs suggested that MexB alone adopts an asymmetric state, in which one of the protomer is in a closed inactive conformation. In contrast, MexB assembled into MexAB-OprM complex adopts an asymmetric functional state, similar to that described before for AcrAB-TolC.

The closed inactive conformation was not observed before in these transporters. The finding of this conformation is novel, albeit it requires further validation. However, the bulk of the results is largely confirmatory.

The structure of MexB is resolved at 4.6 Å, which is quite low and raises concerns about possible overinterpretation. This novel closed conformation requires further validation. For example, mutagenesis stabilizing the closed MexB state and functional characterization of the mutants including an investigation how such mutations affect the assembly and stability of the complex are essential. Without such validation, the result is open to a possibility that this conformation is an artifact.

The reviewer has a concern about the possible overinterpretation of MexB-solo structure. As this concern is already raised by Reviewer 1, we have already answered this question by providing additional material supporting that the quality of the cryoEM map is good enough to build an accurate atomic model.

We would like to mention that the advantage of this experiment avoids the bias of analysing the structure of MexB in a different condition than that of the tripartite structure. Both proteins have been submitted to the same protocol.

Regarding the Reviewer's suggestion of mutagenesis stabilizing the closed state, we agree that such an approach has been extensively used to reinforce our understanding of drug transport. As a proof of this, Pos and collaborators in a recent review (Kobylka *et al.*, 2020) summarize the mutation analysis of the gene *acrB*. Among the 1049 amino acids constituting AcrB, 235 were exchanged and were located in all the three main domains *i.e.* transmembrane domain, the pore domain and funnel domain contributing to understand the drug transport including identification of recognition sites (binding pockets). According to MD simulation on AcrB, the exit gate located in the pore domain engages R124 and Q125 of a small α -helix that moves away from Y758 thereby opening a passage for the substrate. Unfortunately, no mutation has been described for these residues.

If exchanging one of these amino-acids is technically feasible, it seems very unlikely that such mutation could prevent the ad-hoc movement of the helix. In order to support this claim, we have decided to compute a morphing depicting the overall conformational changes occurring when the protein undergoes a transition from MexB-solo to MexB-trio (additional material). This calculation highlights that several subdomains move sequentially and substantially starting from protomer T and transmitted to the neighbouring protomer. So the helix shift will certainly occur despite the substitution of amino acid residues and finally no relevant clue about this transition will be obtained.

In spite of the latter caveat, if a sensible mutation were to be identified, we would reason as follows. In this cascade of conformational changes, the starting point is probably a key point for mutation. In the manuscript, we point out that the Membrane proximal (MP) domain of MexA-II contact PN2 domain of MexB. Q319 of protomer T has a unique contact with MexA involving Q330 and P331. And we make the hypothesis that this contact triggers the cascade of conformational changes. This Q319 residue corresponds to S319 for AcrB. Fujihira *et al.* (2002) have shown that S319C mutation induces a loss of resistance to novobiocin. This mutation hampers the functioning of the pump and correlates well with our hypothesis that the correct positioning of MexA MP domain on MexB is of importance for the pump activity.

From our study, we highlight that the MP domain of MexA-II contributes in maintaining the asymmetric LTO conformation of MexB which corresponds to the functional conformation. The role of allosteric factors for MexA-OprM complex is in good agreement with the recent conclusions of MD simulations performed by Jewel *et al.*, 2020 (detailed explanations are presented in the response to Reviewer1 concerning dynamics information).

We have the following sentence at the end of the paragraph “**Rigid scaffold of MexA induces LTO conformation of MexB**”

It is of interest to note that the mutation of the equivalent residue in AcrB (S319C) resulted in a loss of resistance to novobiocin (Fujihira *et al.* 2002).

The mechanism of the activation remains unclear. There are numerous *in vivo* and *in vitro* studies demonstrating that an MFP and a transporter form a complex even in the absence of an outer membrane channel. In contrast, the authors claim that OprM is required for MexA to bind MexB. This needs further experiments to demonstrate.

We do not claim that MexA needs OprM to bind to MexB. Instead of that, according to transport activity measurement, we observe a gradual increase of proton transport first in the presence of MexA and then in the presence of both MexA and OprM. These results indicate

that MexA definitively binds to MexB. But a fully active transport requires the presence of OprM. Therefore, we suggest that when OprM binds to MexA, it induces the correct positioning of MexA domains on MexB. An argument supporting this assertion is provided by the CusBA structure which is the only bipartite structure available. As explained in the ms page 10” *the positioning of CusB-II on CusA is different from that of MexA-II on MexB and more importantly CusB trimer is symmetric indicating that three partners are required to get a full activity of the RND transporter*”.

Although a "sequential" switching from a closed to an open conformation is proposed, the sequence of events remains unclear.

Tsutsumi *et al.* (2019) have provided structural details of OprM-MexA complex and proposed a sequential event leading to OprM opening. While this study details the upper part of the pump, we focus on the lower part and elaborate on the events leading to LTO conformations of MexB.

Actually, two gates are opened during the assembly process, one corresponding to the opening of the periplasmic helices of OprM and one corresponding to the opening of MexB *i.e.* protomer O. Based on our data (functional and structural) and others including bipartite CusBA, tripartite MexAB-OprM (Tsutsumi *et al.* and ours) and AcrAB-TolC (Wang *et al.*, 2017), we propose a mechanism for MexAB-OprM complex formation that pushes further the one proposed by Tsutsumi *et al.* (2019). An intermediate MexA-MexB complex is formed. OprM engages contacts with MexA molecules resulting in the formation of an open channel made of a hexamer of MexA facing the six helix-turn helix of OprM. By the adjunction of OprM on the bipartite MexAB, OprM and MexA form a corset-like platform that induces the switch from a closed to open conformation of MexB, leading to the asymmetric LTO conformation of MexB in the tripartite complex. We added a movie highlighting these movements as supplemental material to illustrate the suggested functioning.

Reviewer #3 (Remarks to the Author):

This manuscript provides new structural insight into the tripartite multidrug efflux system MexAB-OprM based on nanodisc-reconstituted components and cryo-EM. By analyzing the membrane-embedded MexB trimer before and after assembling the tripartite complex, the authors identify a key role of MexA and OprM in setting the transporter into a functional state. A new stop flow approach demonstrates the *in vitro* proton transport activity of the fully assembled MexAB-OprM complex.

The topic explored here is of great interest and the claims are certainly novel and potentially impactful. I also appreciate the combination of biochemical data with structural data. However, the design and the key data sets for the *in vitro* activity assay have been published before by the authors (Ref 22, Verchère *et al.* 2015, Nat Commun 6, 6890). The new assay presented here has a higher time resolution but the data shown do not add new information. In fact it remains unclear if the MexAB-OprM complex is active in POPC used for the preparation of the nanodiscs samples, since proteoliposomes composed of DOPC and DOPC/cholesterol were used in the transport assay. The novel assay would allow testing this issue and thereby supporting the relevance of the presented structures based on POPC nanodiscs.

We thank the reviewer for acknowledging the relevance of our approach. Indeed, the activity assay has already been published before but we would like to stress that adapting it to pre-steady state kinetic measurement required significant improvements (see below) and made it possible to attain original, sensible and useful information.

As noted already in Verchère *et al.* (2015), the acceleration of acidification (as well as substrate transport) occurs very shortly after proton gradient is generated. In the 2015 paper, everything happened during a “blind window” of few seconds necessary to acidify the fluorescence cuvette in a standard fluorimeter. What we show here is that what we anticipated to be a fast event, turns out to be an *extremely fast* event: with a $t_{1/2}$ of about 100 msec. We believe that this is an important information reflecting that MexB is a rapid and efficient machine (provided that it is assembled with its partners). Of course it will be very desirable to be more quantitative and, for example, suggest stoichiometries of transport. This goal is now at reach but will require additional efforts in the quantification of proteins per liposomes and in the measurement of substrate transport. We believe that, as such, the biochemical data presented here bear all the relevant information necessary to support and explain our structural finding.

Lipid membranes are known to be leaky to protons and our initial rationale was that acidification of liposomes would occur faster in the presence of an active MexB in the membranes. In other words, we previously measured an acceleration of the acidification upon substrate transport. This approach was not relevant for a precise measurement of proton specifically transported by the protein. Hence, undertaking stopped flow measurements required major optimisations in the lipid composition of the liposome so that passive leakage of proton would be restricted to minimum. We followed a fine-tuning of the pH gradient necessary to activate the system versus the tightness threshold of the liposome. From this point of view, the use of POPC instead of DOPC would appear suboptimal as they are known to provide more proton leakage (see e.g: Nicol, F. Nir, S. and Szoka, F. C. Effect of Phospholipid Composition on an Amphipathic Peptide-Mediated Pore Formation in Bilayer Vesicles. *Biophysical Journal* **2000**, 78 (2), 818–829.)

To emphasize our claim, we have added:

the following paragraph in the Introduction.

In complement to structural analysis, quantitative measurement of substrate transport is crucial for the detailed understanding of efflux pumps. In 2005, Aires and Nikaido discussed their results regarding the activity of AcrD into proteoliposome and concluded that approximately 0.3 proton are consumed per trimer of AcrD and per second. In another study, EtBr transport by MexB gave rise to an impressive turnover rate of 500 s^{-1} based on a number of pumps previously estimated by immunoblotting methods (Ocaktan *et al.*, 1997; Narita *et al.*, 2003). Hence, the exact order of magnitude of the velocity of transport is still debated. We have decided to address this issue taking advantage of a methodology allowing the monitoring of efflux pumps activity after reconstitution into proteoliposomes and monitoring proton transport using a pH-sensitive fluorescent probe.

And these sentences in the Result section

Previously, we measured the activity of the MexAB OprM pump in a system that lacked the temporal resolution necessary to accurately monitor rates of transport (Verchère *et al.*, 2015).

The acceleration of acidification upon transport happened during a “blind window” of few seconds necessary to acidify the fluorescence cuvette in a standard fluorometer. We have adapted our methodology to stopped-flow fast recording in order to compare transport activity of MexB at several stages of the assembly process of the pump *i.e.* MexB alone (MexB-solo).

And a conclusion sentence at the end of the same paragraph:

Thanks to the adaptation of our methodology to stopped-flow fast recording, we show that proton transport turns out to be an extremely fast event: with a $t_{1/2}$ of about 100 ms, reflecting that MexB is a rapid and efficient machine when assembled with its partners

An additional flaw concerns the Method part that lacks at several locations essential information that should be added at least to the supplement to allow reproduction of the data:

- When was pyranine present during the vesicle reconstitution? Was non-incorporated pyranine removed, how?

Pyranine was present during vesicle reconstitution and non-incorporated probe was removed by use of PD-10 desalting columns.

- The precise emission wavelength and band width for detection of pyranine are not give; at which temperature was the stopped flow assay performed? What was the vesicle concentration?

What was the final meropenem concentration?

Our stopped flow machine is not equipped with a monochromator at the emission. As stated in the text, maximal emission of pyranine (509 nm) is recorded thanks to the use of wide band emission filters centered in the region of pyranine fluorescence (MTO DA 531, obtained from MTO, Massy (France)).

Measurement were performed at 20°C, vesicle concentration was 1mg/ml and [meropenem] was 5µM.

This information is now added in the corrected manuscript.

- Essential data on the final proteoliposome preparations are lacking (which I could also not find in Verchère et al. 2015, Nat Commun 6, 6890): what was the protein/lipid ratio finally in the vesicles, what was the size of the vesicles, how many protein-free vesicles were present?

Vesicle size was checked by dynamic light scattering (Malvern Nanosizer) and Nanoparticle track analysis (NTA, Nanosight NS300 from Malvern Panalytical) and mean diameter was measured to be 110nm. Protein and lipid quantitation were performed, leading to an estimation of 2-3 proteins per liposome. Homogeneity of the suspension was controlled by electron microscopy.

This information is now added in the corrected manuscript.

Considering the low number of protein per liposome, it is difficult to discriminate between proteoliposomes and protein-free vesicles on electron microscopy images. Hence, it would be too speculative to give a fair estimation of the proportion of empty liposomes. Nevertheless, the presence of empty liposomes is not an issue because the acidification background signal

they would generate is negligible compared to the acidification effect performed with proteoliposomes.

We have added the trace of pure liposomes in the Supplementary Fig. 1.

A minor point concerns the use of abbreviations in the abstract that are not explained (LTO, LTC, LTO).

This has been modified accordingly.

REVIEWER COMMENTS

Reviewer #1 (Remarks to the Author):

The authors have provided a detailed rebuttal to the points raised by the reviewers and changed the text of the manuscript. The responses are satisfactory and the revised manuscript is very good. There are a few points that will hopefully be helpful for the authors to consider:

1. The dip in the GS-FSC may be due to the disorder from the nanodisc. Because the phase randomised mask worked very well (no unexpected "bumps" i.e. introduced correlations), the issue isn't the shape of the complex. Indeed, when the soft mask is used (blue line in Supp Fig 2), the FSC starts to fall just before 0.1 1/Å before suddenly rising sharply, which is not seen in the unmasked or phase-randomised mask. This is quite a common problem when a sharp edge mask is used and the object is very close to the mask – the bright pixels at the edge introduce a very strong correlation, leading to an unexpected FSC rise. There is also a difference between a "soft" and a "soft-edge" mask. A "soft" mask follows the shape of the protein whereas a "soft-edge" mask typically applies a Gaussian drop-off in intensity at the edges. The unmasked image (red line, Supp Fig 2) only shows a small "bump", in line with a second "bump" in the masked FSC (blue line). Again, because this doesn't exist in the phase-randomised mask (green line), there is doubt that this is due to the shape of the molecule (see Rosenthal & Rubenstein, 2015 Curr Opin Struct Biol). The MexB trio portion shows a resolution of around 4–5 Å as calculated in the local resolution estimate in Supp Fig 2c. The MexB structure alone is more globular and so there are not any pixels lying at the edge of the image (Supp Fig 5) when a mask is used, and so there is not the same effect of the mask applied as for MexB-trio (compare blue lines Supp Fig 2 and 5).

2. In the introduction new text

"MexB as well as the E coli homologous AcrB occurred as asymmetric homotrimers" might be better as

"MexB as well as the homologous E coli AcrB are asymmetric homotrimers"

3. Line 103 - the following statement attributed to Wang et al. might need some adjustment:

"According to Wang et al.²⁸, the resting state corresponding to the apo form with TolC in closed-state and AcrB in LLL conformation switched to a transport-state in the presence of puromycin in which the AcrB trimer adopted asymmetric LLT, LTT or LTO conformations and TolC is open." Is this what was stated? It might be safer to state

"According to Wang et al.²⁸, the resting state corresponding to the apo form with TolC in closed-state and AcrB in symmetric conformation switched to a transport-state in the presence of puromycin in which the AcrB trimer adopted asymmetric conformations and TolC is open."

4. line 254 " that OprM-MexA interface was stronger than MexA-MexB one" -- needs to be rewritten

5. "Seconde" is still in x-axis of Figure 1.

6. Acidification assay – some remaining questions. Old assay is Suppl Fig 7b (Verchère et al, 2015 Nat Commun) only has control for MexAB-D407N-proteoliposomes and MexAB-D407N-proteoliposomes + meropenem but not for MexBD407N + MexA + OprM/ MexBD407N + MexA + OprM + meropenem as suggested. The previous control also used the old assay, where a large pH drop is observed even for control liposomes or control liposomes + meropenem. However, in the new assay presented on a

much shorter time-scale, there is no pH drop for the controls (MexB, MexB + meropenem, MexA MexB). These controls (MexBD407N + MexA + OprM/ MexBD407N + MexA + OprM + meropenem) would greatly enhance the conclusion given that "By contrast, proton transport measurement of MexB-trio (Fig. 1b) indicated a basal activity in the absence of the substrate and a strongly enhanced one in the presence of substrate" (lines 275-7). Is the conversion between pyranine fluorescence (AU) to pH the same for the new assay with the linear range between pH 6.0 and 7.0?

7. Preparation of MexAB proteoliposomes – it is mentioned that "Homogeneity of the suspension was controlled by electron microscopy" (lines 396-7). The authors could include a micrograph of these liposomes since it has not been shown before in their previous publications on this system.

8. Fig 2 – still difficult reading the conformational changes between MexB-solo and MexB-trio (especially Fig 2a-c). Static pictures can be tricky to show these dynamics (if with added Sup Fig 8 it is difficult to interpret), so the movie is helpful. As a minor point the blue in the "O" doesn't match the blue in the structure.

Reviewer #2 (Remarks to the Author):

The authors addressed the previous criticism constructively. The manuscript is improved.

Reviewer #3 (Remarks to the Author):

In their revised manuscript the authors now address more clearly the new information gained by their stopped-flow measurements. I can follow the arguments and consider the data worth publishing. However, I still see major flaws concerning the Method part to allow reproducibility of the data:

- Which concentration of pyranine was used during reconstitution?
- Data of the DLS measurements and the EM analysis shown be shown.
- Protein and lipid quantification was performed but neither the procedure is explained nor any data are shown. Such measurements are important to characterize the reconstituted system but are far from trivial. Therefore, a clear documentation within the supplemental material is required. It seem that these important information were also missing in the previous publication (Verchère et al. 2015, Nat Commun 6, 6890). Thus, it is impossible to judge on the quality of the statements by the authors.

- On which basis do the author conclude on the number of proteins per vesicle? Also, two types of vesicles were generated. Had both types of vesicles the same characteristics?

These points should be fixed before publication.

In our response below, remarks of the Reviewers are reproduced *in extenso* (in black) and we address their comments and concerns point-by-point (in blue). When necessary, corrections or addendum to the original manuscript have been made: they are reproduced here and inserted in the main text (in red).

Reviewer #1 (Remarks to the Author):

The authors have provided a detailed rebuttal to the points raised by the reviewers and changed the text of the manuscript. The responses are satisfactory and the revised manuscript is very good. There are a few points that will hopefully be helpful for the authors to consider:

1. The dip in the GS-FSC may be due to the disorder from the nanodisc. Because the phase randomised mask worked very well (no unexpected “bumps i.e introduced correlations), the issue isn’t the shape of the complex. Indeed, when the soft mask is used (blue line in Supp Fig 2), the FSC starts to fall just before 0.1 1/Å before suddenly rising sharply, which is not seen in the unmasked or phase-randomised mask. This is quite a common problem when a sharp edge mask is used and the object is very close to the mask – the bright pixels at the edge introduce a very strong correlation, leading to an unexpected FSC rise. There is also a difference between a “soft” and a “soft-edge” mask. A “soft” mask follows the shape of the protein whereas a “soft-edge” mask typically applies a Gaussian drop-off in intensity at the edges. The unmasked image (red line, Supp Fig 2) only shows a small “bump”, in line with a second “bump” in the masked FSC (blue line). Again, because this doesn’t exist in the phase-randomised mask (green line), there is doubt that this is due to the shape of the molecule (see Rosenthal & Rubenstein, 2015 Curr Opin Struct Biol). The MexB trio portion shows a resolution of around 4–5 Å as calculated in the local resolution estimate in Supp Fig2c. The MexB structure alone is more globular and so there are not any pixels lying at the edge of the image (Supp Fig 5) when a mask is used, and so there is not the same effect of the mask applied as for MexB-trio (compare blue lines Supp Fig2 and 5).

We agree with the reviewer comment that the nanodisc disorder induces a dip in the GS-FSC. Following his advice, we have recomputed the two FSC curves presented in supplementary Fig2 with a soft-edge mask that indeed reduces the peak visible before 0.1 1/Å. We have modified the supplementary Figure 2 accordingly.

2. In the introduction new text "MexB as well as the E coli homologous AcrB occurred as asymmetric homotrimers" might be better as

"MexB as well as the homologous E coli AcrB are asymmetric homotrimers"

This has been modified accordingly

3. Line 103 - the following statement attributed to Wang et al. might need some adjustment:

"According to Wang et al.28, the resting state corresponding to the apo form with TolC in closed-state and AcrB in LLL conformation switched to a transport-state in the presence of puromycin in which the AcrB trimer adopted asymmetric LLL, LTT or LTO conformations and TolC is open." Is this what was stated? It might be safer to state

"According to Wang et al.28, the resting state corresponding to the apo form with TolC in closed-state and AcrB in symmetric conformation switched to a transport-state in the presence of puromycin in which the AcrB trimer adopted asymmetric conformations and TolC is open."

This has been modified accordingly

4. line 254 " that OprM-MexA interface was stronger than MexA-MexB one" -- needs to be rewritten

The sentence

"A simplified estimation of the binding energy suggested that OprM-MexA interface was stronger than MexA-MexB one"

has been modified by

"A simplified estimation of the binding energy at the two interfaces suggested a stronger interaction between OprM and MexA than between MexA and MexB."

5. "Seconde" is still in x-axis of Figure 1.

This has been modified accordingly.

6. Acidification assay – some remaining questions. Old assay is Suppl Fig 7b (Verchère et al, 2015 Nat Commun) only has control for MexAB-D407N-proteoliposomes and MexAB-D407N-proteoliposomes + meropenem but not for MexBD407N + MexA + OprM/ MexBD407N + MexA + OprM + meropenem as suggested. The previous control also used the old assay, where a large pH drop is observed even for control liposomes or control liposomes + meropenem. However, in the new assay presented on a much shorter time-scale, there is no pH drop for the controls (MexB, MexB + meropenem, MexA MexB). These controls (MexBD407N + MexA + OprM/ MexBD407N + MexA + OprM + meropenem) would greatly enhance the conclusion given that "By contrast, proton transport measurement of MexB-trio (Fig. 1b) indicated a basal activity in the absence of the substrate and a strongly enhanced one in the presence of substrate" (lines 275-7).

The reviewer is right, the controls [MexB_{D407N} + MexA + OprM] versus [MexB_{WT} + MexA + OprM] in the presence or in the absence of substrate were lacking. These negative controls bring additional evidence that the effect is indeed mediated by the activity of the protein. We have performed the experiments suggested by Reviewer 1. They are now appended as a new Panel in supplementary Fig. 1. As anticipated, measurements performed in the presence of the inactive mutant MexB_{D407N} do not show any acidification in the presence of substrate or OprM. It is however to be noticed that the rate of acidification shown for the wild type proteins is different from that shown in Figure 1. It is standard observation, in our hands, that the rates of acidification differ from batch to batch, most certainly because the maximal rate of acidification depends on different factors such as the yield of protein reconstituted per liposome or the efficiency of assembly between MexAB proteoliposomes and OprM proteoliposomes.

Is the conversion between pyranine fluorescence (AU) to pH the same for the new assay with the linear range between pH 6.0 and 7.0?

Up to now, we typically standardized pyranine fluorescence as a function of pH on the basis of a theoretical pyranine concentration (namely that used during the reconstitution process, 5mM), although the actual quantity indeed entrapped is most certainly much lower. During the course of the reviewing process, we realized that it would be more relevant to experimentally check and measure fluorescence linearity for the pyranine quantity trapped in the liposome lumen. Hence, as we undertook a new set of experiments to address Reviewers'

comments, we dedicated one third of the proteoliposome preparation to check that pyranine fluorescence changes were also linear between pH 6 and pH 7.5 for concentrations as low as that entrapped in the liposomes' lumen. Emitted pyranine fluorescence was measured using excitation wavelengths ranging from 452 to 460 nm on proteoliposomes resuspended in buffers with decreasing pH. Measurements were made in the presence of 10 μ M valinomycin and 10 μ M nigericin to facilitate transmembrane proton equilibration. As can be seen in a new Supplementary Fig. 11, the fluorescence variations are perfectly linear between pH 5 and pH 7.5.

Accordingly, the following sentence has been added in Material and Method section, paragraph Stopped flow measurements

“Linearity of pyranine fluorescence over the pH range has been assessed (Supplementary Fig. 11).”

7. Preparation of MexAB proteoliposomes – it is mentioned that “Homogeneity of the suspension was controlled by electron microscopy” (lines 396-7). The authors could include a micrograph of these liposomes since it has not been shown before in their previous publications on this system.

Following the reviewer request, we added two micrographs of the MexAB proteoliposomes obtained by negative-staining electron microscopy and cryo-electron microscopy (Supplementary Fig. 10). CryoEM observation reveals that the proteoliposomes have a spherical shape. Negative-staining EM micrograph shows densities protruding from the liposomes likely corresponding to MexB trimers. It was difficult to observe MexA molecules around MexB as MexA is more flexible. From cryo-EM observations, proteoliposomes have a size in a range of 50-150 nm. A size measurement in solution using dynamic light scattering indicates that most of the proteoliposomes have a size average of about 110 nm.

Supplementary Fig. 10 is mentioned in the paragraph “Preparation of MexAB proteoliposomes (Material and Method section).

8. Fig 2 – still difficult reading the conformational changes between MexB-solo and MexB-trio (especially Fig 2a-c). Static pictures can be tricky to show these dynamics (if with added Sup Fig 8 it is difficult to interpret), so the movie is helpful. As a minor point the blue in the “O” doesn't match the blue in the structure.

To take into account the remark of the reviewer, additional panels have been added in the figure 2 For the better visualisation of the conformation changes

The old figure 2

is replaced by the new figure 2. The legend is modified accordingly

Fig 2: Structure of MexB-solo and MexB-trio

a, Pore domain visualization of MexB-solo trimer in LTC conformation shown in yellow, orange and red for the L, T and C protomers respectively. **b-c**, Superimposition of MexB-solo and MexB-trio (purple, green and blue for the L, T and O protomers respectively) viewed from the top (b) as in (a) and perpendicular to the membrane (c). In the periplasmic part, main

conformational changes observed between MexB-solo and MexB-trio occurred in the three β -hairpin loops (DN domain) (c), in the T protomer PN2 domain, and in the neighboring protomer switched from C to O state with a combined movement of the PC2 domain and the central helix plus helix gate (PN1 domain). The described movements are illustrated by red arrows. **d**, Pore domain visualization of MexB-trio trimer. The conformational changes observed in (b-c) between MexB-solo and MexB-trio are shown in purple, green and blue (for the L, T and O protomers respectively), according to the root mean square deviations. Parts of the structure with rmsd lower than 2 Å are in grey. The PN2 and PC2 shifts toward their respective PC1 domain are indicated by curved dashed black arrows. Note the colored β -hairpin loops illustrating their conformational changes (red asterisks). **e**, Surface representation of tripartite complex at the level of pore domain of MexB. The three MP domains of MexA-I are contoured. The red arrow indicates the MP domain in contact with the portion of T protomer undergoing the large rmsd corresponding to the “green” segments shown in (d). MexA-I and MexA-II models are shown in light-green and cyan respectively. **f**, View of the exit gate delineated by the helix gate E121-R124 on the left and the β strand on the right in protomer C of MexB-solo (red) and in protomer O of MexB-trio (cyan). The helix in protomer O of MexB-trio moves away forming a passage for drug exit that is hampered in protomer C of MexB-solo as shown with the tunnel (blue) calculated using CAVER.

Reviewer #2 (Remarks to the Author):

The authors addressed the previous criticism constructively. The manuscript is improved.

We appreciate the positive comment.

Reviewer #3 (Remarks to the Author):

In their revised manuscript the authors now address more clearly the new information gained by their stopped-flow measurements. I can follow the arguments and consider the data worth publishing.

We thank the reviewer for his/her positive comments.

However, I still see major flaws concerning the Method part to allow reproducibility of the data:

- Which concentration of pyranine was used during reconstitution?

The concentration of pyranine used during reconstitution is 5 mM but the actual quantity indeed entrapped into the liposome lumen is most certainly much lower. Fluorescence linearity was checked for the pyranine quantity indeed trapped in the liposome lumen. Emitted pyranine fluorescence was measured using excitation wavelengths ranging from 452 to 460 nm on proteoliposomes resuspended in buffers with decreasing pH. Measurements were made in the presence of 10 μ M valinomycin and 10 μ M nigericin to facilitate transmembrane proton equilibration. As presented now in supplementary Fig. 11, the fluorescence variations are perfectly linear between pH 5 and pH 7.5.

- Data of the DLS measurements and the EM analysis shown be shown.

As the EM analysis have also been asked by the Reviewer #1, we responded to his/her comment above by showing the micrograph for MexAB proteoliposomes.

We added the DLS measurements analysis (Supplementary Fig. 10). For each proteoliposome, measurement has been repeated three times for each sample. Average sizes and polydispersity are summarized in Supplementary Fig. 10c. From these results, proteoliposome sizes are around 110 nm for MexAB proteoliposomes and 130 nm for OprM proteoliposomes, with a polydispersity below 0.2 indicating that samples are homogeneous.

- Protein and lipid quantification was performed but neither the procedure is explained nor any data are shown. Such measurements are important to characterize the reconstituted system but are far from trivial. Therefore, a clear documentation within the supplemental material is required. It seems that these important information were also missing in the previous publication (Verchère et al. 2015, Nat Commun 6, 6890). Thus, it is impossible to judge on the quality of the statements by the authors.

- On which basis do the author conclude on the number of proteins per vesicle? Also, two types of vesicles were generated. Had both types of vesicles the same characteristics?

In order to approximate the amount of proteins incorporated into liposomes, the amount of lipids has been estimated by quantitatively determining lipid phosphorus with a linear range after the phosphorus is converted to inorganic phosphate by means of a perchlorate digestion. The stable complex was read with a spectrophotometer at a wavelength of 797 nm. Based on a standard curve, a quantity of 29 nmol of lipids has been determined for 15 μ L of proteoliposomes (1). Assuming an average proteoliposome diameter of 110 nm according DLS measurement and a surface area per lipid molecule to be 0.70 nm², the average number of lipid molecules per proteoliposome is estimated to be 99,185 (2). From (1) and (2), there were about 1.76 10¹¹ liposomes per 15 μ L of solution (3). Then the amount of proteins has been estimated on SDS-PAGE gel, using the Image Lab software from a range of purified protein. A 15 μ L of proteoliposomes contained 271 \pm 6 ng of MexA and 165 \pm 20 ng of MexB corresponding to 40.7 10¹¹ MexA monomer and 8.79 10¹¹ MexB monomer (4). From (3) and (4) we estimated that our reconstitution procedure gives rise to 23 MexA and 5 MexB monomers (i.e 1.6 MexB trimers) per a 110 nm liposome.

The previous sentence in “Preparation of MexAB proteoliposomes” paragraph (Material and Method section)

“ Protein and lipid quantitation were performed, leading to an estimation of 2-3 proteins per liposome”

is now replaced by the following sentences:

“In order to approximate the amount of proteins incorporated into liposomes, the amount of lipids has been estimated by quantitatively determining lipid phosphorus with a linear range after the phosphorus is converted to inorganic phosphate by means of a perchlorate digestion. The stable complex was read with a spectrophotometer at a wavelength of 797 nm. Based on a standard curve, a quantity of 29 nmol of lipids has been determined for 15 μ L of proteoliposomes (1). Assuming an average proteoliposome diameter of 110 nm according DLS measurement and a surface area per lipid molecule to be 0.70 nm², the average number of lipid molecules per proteoliposome is estimated to be 99,185 (2). From (1) and (2), there were about 1.76 10¹¹ liposomes per 15 μ L of solution (3). Then the amount of proteins has been estimated on SDS-PAGE gel, using the Image Lab software from a range of purified protein. A 15 μ L of proteoliposomes contained 271 \pm 6 ng of MexA and 165 \pm 20 ng of MexB corresponding to 40.7 10¹¹ MexA monomer and 8.79 10¹¹ MexB monomer (4). From (3) and (4) we estimated that our reconstitution procedure gives rise to 23 MexA and 5 MexB monomers (i.e 1.6 MexB trimers) per a 110 nm liposome.

These points should be fixed before publication.

REVIEWERS' COMMENTS

Reviewer #1 (Remarks to the Author):

The authors have provided detailed and compelling responses to all the reviewer comments. The manuscript is improved with the additional changes.

Reviewer #3 (Remarks to the Author):

The authors have addressed most of my comments adequately and I can recommend the publication of the manuscript.

Some small points in the method section should be fixed before publication to enhance the reproducibility of the data:

(1)Line 426/427...and subjected to 5 rounds of sonication for 30 seconds each. Lipid concentration was quantified by phosphate analysis.

Please specify the type of sonication: tip sonicator/bath sonicator, power settings, duty cycl, glss tubes? Eppendorfs? etc.

Please add reference or specify the procedure of the phosphate assay.

(2)Please specify how was Negative-staining EM performed.

In our response below, remarks of the Reviewers are reproduced *in extenso* (in black) and we address their comments and concerns point-by-point (in blue). When necessary, corrections or addendum to the original manuscript have been made: they are reproduced here and inserted in the main text (in red).

Reviewer #1 (Remarks to the Author):

The authors have provided detailed and compelling responses to all the reviewer comments. The manuscript is improved with the additional changes.

We appreciate the positive comment.

Reviewer #3 (Remarks to the Author):

The authors have addressed most of my comments adequately and I can recommend the publication of the manuscript.

We appreciate the positive comment.

Some small points in the method section should be fixed before publication to enhance the reproducibility of the data:

(1)Line 426/427...and subjected to 5 rounds of sonication for 30 seconds each. Lipid concentration was quantified by phosphate analysis.

Please specify the type of sonication: tip sonicator/bath sonicator, power settings, duty cycl, glss tubes? Eppendorfs? etc.

The following information is added in the material section.

(sonicator tip, 30 sec pulse, 30 sec pause in a glass tube) at 4 Watts

Please add reference or specify the procedure of the phosphate assay.

The following reference is added:

Rouser, G., Fleischer, S. & Yamamoto, A. Two dimensional thin layer chromatographic separation of polar lipids and determination of phospholipids by phosphorus analysis of spots. *Lipids* **5**, 494–496 (1970).

(2)Please specify how was Negative-staining EM performed.

The following information is added in the material section.

Sample was deposited on a glow-discharged carbon-coated copper 300 mesh grids and stained with 2% uranyl acetate (wt/vol) solution. Images were acquired on a Tecnai F20 electron microscope (ThermoFisher Scientific) operated at 200kV using an Eagle 4k_4k camera (ThermoFisher Scientific)